Journal of Data-centric Machine Learning Research (2024)        Submitted 12/23; Revised 2/24; Published 3/24

# LabelBench: A Comprehensive Framework for Benchmarking Adaptive Label-Efficient Learning

**Jifan Zhang**[*]                                                    JIFAN@CS.WISC.EDU
*University of Wisconsin, Madison, WI*

**Yifang Chen**[*]                                          YIFANGC@CS.WASHINGTON.EDU
*University of Washington, Seattle, WA*

**Gregory Canal**                                                   GCANAL@WISC.EDU
*University of Wisconsin, Madison, WI*

**Arnav M. Das**[†]                                                 ARNAVMD2@UW.EDU
*University of Washington, Seattle, WA*

**Gantavya Bhatt**[†]                                               GBHATT2@UW.EDU
*University of Washington, Seattle, WA*

**Stephen Mussmann**                                          MUSSMANN@GATECH.EDU
*Georgia Institute of Technology, Atlanta, GA*

**Yinglun Zhu**                                                      YZHU@UCR.EDU
*University of California, Riverside, CA*

**Jeffrey Bilmes**                                                  BILMES@UW.EDU
*University of Washington, Seattle, WA*

**Simon S. Du**                                             SSDU@CS.WASHINGTON.EDU
*University of Washington, Seattle, WA*

**Kevin Jamieson**                                       JAMIESON@CS.WASHINGTON.EDU
*University of Washington, Seattle, WA*

**Robert D. Nowak**                                                RDNOWAK@WISC.EDU
*University of Wisconsin, Madison, WI*

**Reviewed on OpenReview:** *https://openreview.net/forum?id=Y2QcZfwHE7*

**Editor:** Yue Zhao

## Abstract

Labeled data are critical to modern machine learning applications, but obtaining labels can be expensive. To mitigate this cost, machine learning methods, such as transfer learning, semi-supervised learning and active learning, aim to be *label-efficient*: achieving high predictive performance from relatively few labeled examples. While obtaining the best label-efficiency in practice often requires combinations of these techniques, existing benchmark and evaluation frameworks do not capture a concerted combination of all such techniques. This paper addresses this deficiency by introducing LabelBench, a new computationally-efficient framework for joint evaluation of multiple label-efficient learning techniques. As

---

[*] Equal contribution.
[†] Equal contribution.

an application of LabelBench, we introduce a novel benchmark of state-of-the-art active learning methods in combination with semi-supervised learning for fine-tuning pretrained vision transformers. Our benchmark demonstrates significantly better label-efficiencies than previously reported in active learning. LabelBench's modular codebase is open-sourced for the broader community to contribute label-efficient learning methods and benchmarks. The repository can be found at: `https://github.com/EfficientTraining/LabelBench`.

**Keywords:** Label-Efficient Learning, Large Pretrained Model

## 1 Introduction

Large pretrained models provide practitioners with strong starting points in developing machine-learning-powered applications (Radford et al., 2021; Yu et al., 2022; Kirillov et al., 2023). While zero-shot and few-shot predictions can provide solid baselines, linear probing (which freezes the model and trains a layer on top) and fine-tuning based on human annotation yield significantly better performance (Radford et al., 2021; Yu et al., 2022). Label-efficient learning, which aims to achieve high predictive performance with fewer labels, has received much attention lately due to the high annotation cost of labeling large-scale datasets.

Transfer learning, semi-supervised learning (Semi-SL) and active learning (AL) all study different aspects of label-efficient learning. Modern transfer learning leverages large general-purpose models pretrained on web-scale data and fine-tunes the model to fit application-specific examples. Semi-supervised learning utilizes a large set of unlabeled examples to estimate the underlying data distribution and more efficiently learn a good model. Active learning incrementally and adaptively annotates only those examples deemed to be informative by the model. To date, however, no existing literature has studied all the above methods under a single unified benchmarking framework for fine-tuning large pretrained models.

In this paper, we present LabelBench, a comprehensive benchmarking framework for *label-efficient* learning. Our framework tackles computational challenges that arise when scaling these techniques to large neural network architectures. Specifically, incorporating AL involves periodically retraining the model based on the latest labeled examples. While repeatedly training small convolutional neural networks is practically feasible (Sener and Savarese, 2017; Ash et al., 2019, 2021; Beck et al., 2021; Zhan et al., 2022; Lüth et al., 2023), retraining large-scale models is extremely compute intensive, making large scale AL experiments prohibitively expensive (Das et al., 2023). Inspired by selection-via-proxy (Coleman et al., 2019), we propose lightweight retraining schemes (based on freezing all but the last layer of large pretrained models) for the purpose of data selection and labeling, but evaluate final model performance with a single end-to-end fine-tuning. This technique yields a ten-fold reduction in training cost, and reaps most of the label-efficiency gains of using AL.

To showcase the power of our framework, we conduct experiments that benchmark multiple deep active learning algorithms in combination with semi-supervised learning and large pretrained models; our experiments reveal especially strong label-efficiency gains from active learning, demonstrating a significant difference from conventional beliefs in existing literature. To highlight some of our results, we observe a more than four-fold reduction (75% savings) in annotation cost over random sampling on CIFAR-10 (Figure 1(a)), a dataset

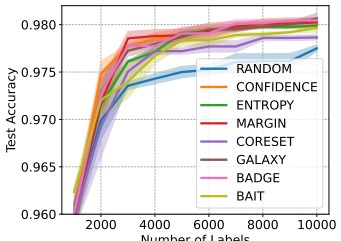
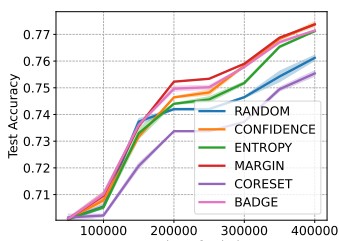
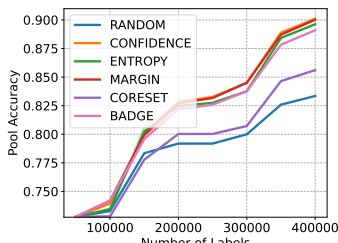

(a) Generalization Acc, CIFAR-10  (b) Generalization Acc, Ima-geNet  (c) Pool Accuracy on ImageNet

Figure 1: Performance of active learning + FlexMatch (semi-supervised) retraining + CLIP ViT-B32 when given different annotation budgets. Generalization accuracy refers to the model's Top-1 test accuracy. Pool accuracy measures the labeling accuracy on the pool of examples to be labeled (see Section 4.2 for more details). Each curve of CIFAR-10 is averaged over 4 trials and each curve of ImageNet is averaged over two trials. The confidence intervals are based on standard error. The AL gains over passive presented here are significantly larger than typical gains observed in previous AL work where Semi-SL and pretrained models are not considered.

that was believed to be particularly challenging for active learning [*]. This improvement is further demonstrated in our experiments on ImageNet (Figure 1(b, c)). Under any fixed annotation budget, our experiments suggest active learning algorithms can consistently boost test accuracy by more than 1.2% and pool accuracy (accuracy of predictions on the pool of unlabeled training data, as defined in Section 4.2) by more than 5%. Compared to the previous best results in this setting (Emam et al., 2021), our results yield at least 10% higher test accuracy. Overall, LabelBench provides a light-weight benchmarking framework for researchers to test their algorithms on more realistic and large-scale scenarios.

## 2 Related Work

Large pretrained models have demonstrated a wide range of generalization abilities on downstream language and vision tasks. Most of these models are trained on web-scale data with supervised (Kolesnikov et al., 2019; Dosovitskiy et al., 2020; Zhai et al., 2021) or self-supervised techniques (Radford et al., 2021; Jia et al., 2021; Yuan et al., 2021; Singh et al., 2021; Yao et al., 2021; Wang et al., 2022a; Yu et al., 2022). While these models are powerful by themselves, adapting them to applications often requires transfer learning by fine-tuning on human annotated examples. Below we survey existing literature on label-efficient learning with an emphasis on the interplay among large pretrained models, semi-supervised learning and active learning.

---

[*]As reported in seminal papers and common benchmarks such as Ash et al. (2019) (Figure 16), Ash et al. (2021) (Figure 10), Beck et al. (2021) (Figure 1) and Lüth et al. (2023) (Figure 6-8), they see less than two-fold reductions in annotation cost.

## 2.1 Semi-supervised Training

In traditional supervised learning the model is only trained on the set of *labeled* examples, while in Semi-SL the model training is also informed by the remaining *unlabeled* examples in the pool. Intuitively, Semi-SL leverages the assumption that examples lying "nearby" to one another should belong to the same class, and therefore during training the model is encouraged to produce the same model output for these examples (for an overview of Semi-SL we refer the interested reader to Zhu (2005); van Engelen and Hoos (2020); Ouali et al. (2020)). Broadly speaking, modern Semi-SL methods implement this principle using a combination of *Consistency Regularization* — where model outputs of neighboring examples are regularized to be similar — and/or *Pseudo Labeling* — where unlabeled examples that the model is confident on are assigned artificial labels to supplement supervised training (Sohn et al., 2020; Berthelot et al., 2020). In our pipeline, we implement Pseudolabeling (Lee, 2013), Unsupervised Data Augmentation (Xie et al., 2020b), FlexMatch (Zhang et al., 2021a), FreeMatch (Wang et al., 2022c) and SoftMatch (Chen et al., 2023).

**Semi-supervised Training of Large Pretrained Models.** The application of Semi-SL to fine-tuning large pretrained models is a nascent area of research. Cai et al. (2022) pioneered the application of Semi-SL methods to large-scale vision transformers by using a multi-stage pipeline of pretraining followed by supervised fine-tuning and finally semi-supervised fine-tuning. Lagunas et al. (2023) apply this pipeline to a fine-grained classification e-commerce task and demonstrate improved performance compared to standard supervised training. Semi-SL training on transformer architectures has also been successfully applied to video action recognition (Xing et al., 2023). USB (Wang et al., 2022b) is a benchmark that includes Semi-SL evaluations on large pretrained models such as ViT; however, it does not incorporate AL into its pipeline, as we do here.

## 2.2 Active Learning

If we have a large pool of unlabeled examples and a limited labeling budget, one must select a subset of the data for label annotation. Various strategies have been proposed to identify an informative subset that produces a good model from a limited budget of labels. Experimental design (Pukelsheim, 2006) studies the setting where this subset is chosen before any annotations are observed. Pool-based active learning (Settles, 2009) examines iterative adaptive annotation: labels from previously annotated examples can be used to determine which examples to choose for annotation in the next iteration. Active learning algorithms are generally designed to maximize one or both of the intuitive concepts of *uncertainty* and *diversity*. Uncertainty, measured in a variety of ways (Settles, 2009), refers to the uncertainty of a trained model for the label of a given point (Lewis, 1995; Scheffer et al., 2001), while diversity refers to selecting points with different properties (Sener and Savarese, 2017). Many algorithms maximize a combination of these two concepts (Ash et al., 2019; Wei et al., 2015; Ash et al., 2021; Citovsky et al., 2021; Zhang et al., 2022).

**Active Learning for Fine-Tuning Large Pretrained Models.** Recent literature in deep active learning has started to utilize large pretrained models for large-scale datasets. Coleman et al. (2022) proposes a computationally efficient method to annotate billion-scale datasets by actively labeling examples only in the neighborhood of labeled examples in the SimCLR (Chen et al., 2020) embedding space. Tamkin et al. (2022) found that

uncertainty sampling yields larger annotation saving for large pretrained models than for traditional ResNet. LabelBench serves as a more comprehensive large-scale benchmark for these studies, where we combine Semi-SL training in our framework. We further take into account the expensive cost of fine-tuning large pretrained models at every iteration of active data collection.

In addition, numerous papers have utilized self-supervised or unsupervised learning methods to initialize their models (Siméoni et al., 2021; Chan et al., 2021; Wen et al., 2022; Lüth et al., 2023) on the unlabeled datasets. However, their methods do not utilize existing large pretrained models.

**Active Learning with Semi-supervised Training.** Since AL and Semi-SL seek to maximize model performance using only a minimal budget of labeled points, it is natural to combine both techniques (Guillory and Bilmes, 2011) to maximize label efficiency. This practice dates back to Zhu et al. (2003), which labels examples that minimize expected classification error in a Gaussian Field Semi-SL model. In the context of deep learning, Lüth et al. (2023); Chan et al. (2021); Mittal et al. (2019); Siméoni et al. (2021) benchmark various AL methods in Semi-SL settings. Huang et al. (2021) develops a hybrid AL/Semi-SL approach for computer vision tasks, and Gao et al. (2020) develops a consistency-based AL selection strategy that is naturally compatible with Semi-SL methods. Borsos et al. (2021) approaches AL in the context of Semi-SL as a problem of dataset summarization, and demonstrates improved performance on keyword detection tasks. Hacohen et al. (2022); Yehuda et al. (2022) both use FlexMatch as a baseline Semi-SL method in their AL experiments, further corroborating our choice of FlexMatch in our own pipeline.

## 3 Label Efficient Fine-tuning Framework

We propose a framework for label-efficient learning consisting of three widely-adopted components in modern deep learning: initialization with a large pretrained model, data annotation, and fine-tuning on downstream tasks. Our framework supports traditional AL, but also takes advantage of large pretrained models and Semi-SL to further improve the label-efficiency. As shown in Figure 2, our framework starts with a large pretrained model as initialization. Data annotation follows a closed-loop procedure, where one starts with a pool of unlabeled examples in the beginning and iteratively gathers more human annotations. At any iteration, given a partially labeled pool we utilize semi-supervised training to obtain the best performing model. Informed by this trained model, an active learning strategy selects unlabeled examples it deems the most informative and sends those examples to be labeled. At the end of the iteration, the newly annotated labels are recorded into the dataset.

The greatest challenge in implementing this framework comes from incorporating large-scale model training while meeting a limited computational budget. Unlike past deep active learning methods (Sener and Savarese, 2017; Ash et al., 2019, 2021) that utilize smaller neural network architectures (e.g., ResNet-18), the computational cost of fine-tuning large pretrained models at every iteration of the data collection loop is a significant burden. To address this challenge, we propose using a *selection-via-proxy* (Coleman et al., 2019) approach (Section 3.1), along with additional code optimization to improve the computational and memory efficiencies for large-scale datasets. In addition, our codebase is modular, allowing contributors to easily work on isolated components of the framework (Section 3.2).

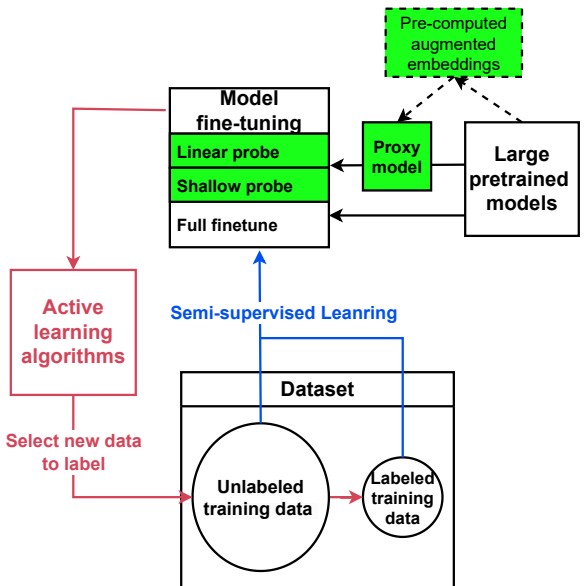

```
# Add a new dataset.
@register_dataset(
    "my_dataset", MULTI_CLASS)
def get_dataset(...):
    ...

# Add a new Semi-SL Algorithm.
Class MyTrainer(SemiTrainer):
    def train_step(img,
                   aug_img,
                   ...):
        ...
```

Figure 3: Our modular codebase allows one to work solely in one directory without a thorough knowledge of the entire codebase. Implementing a new dataset or semi-supervised learning trainer is as easy as implementing a single function.

Figure 2: A modular framework consisting of pretrained models, Semi-SL trainer and AL strategies.

|  | End-to-end Fine-Tune | | Shallow Network(proxy) | |
| --- | --- | --- | --- | --- |
| Training Stage | GPU Hours | AWS Dollars | GPU Hours | AWS Dollars |
| Precomputation | 0 | $0 | 5 | $15 |
| Retraining | 1900 | $5700 | 57 | $180 |
| Final Model | 100 | $300 | 100 | $300 |
| **Total** | 2000 | $6300 | 162 | $495 |

Table 1: Estimated cost of neural network training for ImageNet experiments when collecting 600,000 labels with 20 iterations (batches of 30,000 labels per iteration). Here we display the total cost of running 12 trials with CLIP ViT-B32 and FlexMatch Semi-SL training (Zhang et al., 2021a). All AWS dollars are based on on-demand rates of EC2 P3 instances.

### 3.1 Selection via Proxy

During each iteration of data collection, there are three potential strategies in fine-tuning the large pretrained model: fine-tuning the model end-to-end, training only a linear probe (Alain and Bengio, 2016), and training a nonlinear probe with a shallow neural network. In the latter two strategies, the learner freezes the pretrained encoder and attaches to it a less computationally intensive model at the output (i.e. a linear classifier or shallow network). This can greatly reduce the computational cost of retraining, but often the final model does not perform as well as one in which the full model is retrained.

To better trade-off between retraining/inference cost and the final model performance, we propose a *selection-via-proxy* approach, which is inspired by Coleman et al. (2019). In the referenced work, a less computationally intensive proxy is created by carefully scaling down the original model architecture and training for fewer epochs. In our framework, we exploit a more straightforward approach by employing the linear probe and shallow network models as potential proxies. During every iteration of the data annotation loop, the learner only retrains the proxy model, which informs the selection of unlabeled examples to be annotated. After collecting a sufficient amount of labeled examples or reaching the labeling budget limits, the learner then switches to end-to-end fine-tuning at the last batch to further boost the performance of the final model. As a result, selection-by-proxy significantly reduces the cost of back-propagation.

We further reduce the forward inference cost by precomputing and saving embeddings of each dataset in advance. To account for random image augmentations during training, we precompute five sets of embeddings on randomly augmented images using different random seeds. Our dataloader loops through these sets of embeddings over different epochs. As shown in Table 1, we highlight the reduction in experimentation cost on the ImageNet dataset. In particular, selection-via-proxy reduces the GPU time and training-induced cost by more than ten-fold.

## 3.2 Codebase

Our codebase consists of five components: datasets, model, training strategy (for supervised and semi-supervised training), active learning strategy and metrics. We would like to highlight the following advantages of our implementation:

- **Modularity.** As shown in Figure 3, adding any new instance, such as a new dataset or training strategy, simply involves implementing a new function. This allows future contributors to solely focus on any isolated component without a thorough understanding of the entire repository.
- **Self-report mechanism.** We include configuration files of all experiment setups. In addition, we keep track of all experiment results in the results directory for fair comparisons. Researchers are encouraged to self-report their research findings by submitting pull requests to our repository.
- **Significant speed-up of existing AL implementation.** Running some AL algorithms can be time-prohibitive when scaled to large datasets with large numbers of classes. In our implementation, we speed up popular active learning algorithms such as BADGE (Ash et al., 2019) and BAIT (Ash et al., 2021) by orders of magnitude in comparison to existing implementations (Appendix E).

## 4 Benchmarking Active Learning Algorithms

To demonstrate the utility of our framework, we conduct experiments comparing popular deep AL strategies in combination with large pretrained models and semi-supervised training. Our results presented in Section 4.4 show significantly better label efficiencies than existing deep AL literature. Moreover, we discuss the accuracy gap by using selection-via-proxy under different settings.

### 4.1 Experiment Setup

Here we detail our benchmark's specific choices of AL strategies, large pretrained models, and Semi-SL methods. It is important to note that settings beyond the ones discussed here can also be easily integrated into our general framework and codebase. We leave details of our hyper-parameter tuning procedure to Appendix D and leave more detailed discussions on potential future directions to Section 5. Our benchmark studies the following annotation procedure:

1. **Initial large pretrained model.** We use pretrained CLIP (Radford et al., 2021) and CoCa (Yu et al., 2022) with the ViT-B32 architecture as image encoders. For end-to-end fine-tuning, we attach the image encoder with a zero-shot prediction linear head. On the other hand, proxy models are initialized with random weights. Throughout our experiments, shallow networks have a single hidden layer with the same dimension as the embeddings.
2. **Initial batch of labels.** We collect the first batch of labels by sampling uniformly at random.
3. **Adaptive annotation loop.** We iterate over the following steps to annotate batches of examples.
   - **Model training.** At the beginning of each iteration, the dataset is partially labeled. We use Semi-SL techniques to fine-tune the vision transformer or train the proxy model from scratch. In particular, we experiment with Semi-SL techniques that minimize a *supervised training loss* on labeled examples and an *unsupervised loss* on unlabeled examples that uses pseudolabeling and/or some form of consistency regularization. Most of our experiments use FlexMatch (Zhang et al., 2021a), but we also experiment with simpler methods such as Unsupervised Data Augmentation (UDA) (Xie et al., 2020b) and Pseudolabeling (Lee, 2013) to assess the sensitivity of our pipeline to the choice of Semi-SL technique.
   - **Data selection.** Given the trained model, we use a data selection strategy to select unlabeled examples for annotation. We benchmark against prevalent active learning algorithms such as confidence sampling (Lewis, 1995), margin sampling (Scheffer et al., 2001), entropy sampling (Settles, 2009), CORESET (Sener and Savarese, 2017), BADGE (Ash et al., 2019) and GALAXY (Zhang et al., 2022) (see Section 2 and Appendix B for details). These algorithms make decisions based on the model's properties and its prediction on the pool of unlabeled examples (e.g. the confidence/entropy score, the gradient of the linear probe).
   - **Annotate.** Based on the strategy's selection, we reveal the true labels and update the dataset.
4. **Final Model.** After the annotation budget is exhausted, regardless if the proxy model is used for selection or not, we fine-tune the pretrained CLIP or CoCa model end-to-end by FlexMatch on the collected labeled examples as well as the remaining unlabeled examples.

### 4.2 Performance Metrics

We report results on the following two tasks of label-efficient learning.
- **Label-efficient generalization** aims to learn accurate models that generalize beyond examples in the pool while spending limited budget on oracle annotation, such as human

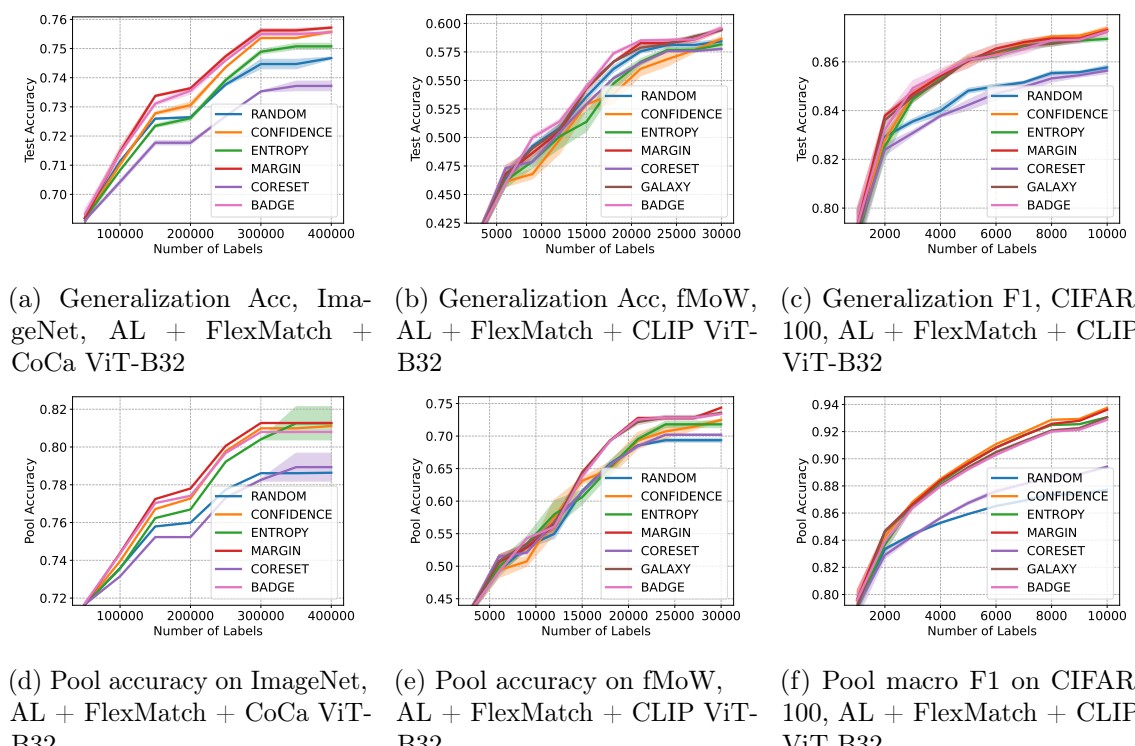

(a) Generalization Acc, ImageNet, AL + FlexMatch + CoCa ViT-B32

(b) Generalization Acc, fMoW, AL + FlexMatch + CLIP ViT-B32

(c) Generalization F1, CIFAR-100, AL + FlexMatch + CLIP ViT-B32

(d) Pool accuracy on ImageNet, AL + FlexMatch + CoCa ViT-B32

(e) Pool accuracy on fMoW, AL + FlexMatch + CLIP ViT-B32

(f) Pool macro F1 on CIFAR-100, AL + FlexMatch + CLIP ViT-B32

Figure 4: Performances of different data selection strategies on ImageNet, fMoW and CIFAR-100. We omit GALAXY in ImageNet due to its expensive computational complexity on large datasets. The ImageNet results differ from Figure 1 since we use a different pretrained model, CoCa ViT-B32. Each result of fMoW and CIFAR-100 is averaged over four trials and each result of ImageNet is over two trials due to limited computing resources. The confidence intervals are based on standard error.

labeling. We refer to the models' performances on test data as *generalization performance*. In this paper, we report performances on in-distribution test data (drawn from the same distribution as the pool). As will be mentioned in Section 5, one may be able to extend this benchmark to distribution shift cases.

- **Label-efficient annotation** aims to annotate all examples in the pool with a limited labeling budget, similar to the objective of transductive learning. When the dataset is partially labeled by a human, a model trained based on existing annotations can serve as a pseudo annotation tool that labels the rest of the unlabeled examples. We refer to the percentage of labels (both human annotated and pseudo labels) that agree with ground-truth labels as the *pool performance*. Examples of label-efficient annotation applications include product cataloging, categorizing existing userbases, etc.

To quantify performance, we use the standard accuracy for (near) balanced datasets, and balanced accuracy and macro F1 score for imbalanced datasets. Balanced accuracy and macro F1 score are measured as unweighted averages of per-class accuracies and per-class F1 scores, respectively.

### 4.3 Datasets

We first test on CIFAR-10, CIFAR-100 (Krizhevsky et al., 2009) and ImageNet (Deng et al., 2009), all of which are standard datasets used in previous AL and Semi-SL papers. To further evaluate LabelBench on more realistic datasets, we also test on iWildCam (Beery et al., 2021) and fMoW (Christie et al., 2018), parts of the WILDS benchmark (Koh et al., 2021). To the best of our knowledge, only a handful of existing studies, such as Tamkin et al. (2022); Mussmann et al. (2022); Bartlett et al. (2022), have evaluated label-efficient algorithms on these datasets, albeit under different experimental setups. The WILDS benchmark was originally intended to represent distribution shifts faced in the wild (i.e., OOD test sets); here we limit our evaluation to in-domain (ID) test set performance as an initial exploratory step. Using these datasets provides several advantages: 1) Both of them are highly imbalanced. 2) Fine-tuning pretrained large-scale models on them is more challenging than on ImageNet (e.g., ID test accuracy on fMoW is 73.3% (Wortsman et al., 2022) when fine-tuning ViT-L14 end-to-end). 3) Unlike ImageNet and CIFAR10, whose examples are gathered by querying search engines with human validation, iWildCam and fMoW gather labels directly from human annotators, which aligns more closely with our pool-based active learning setting.

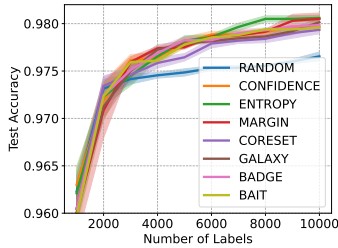 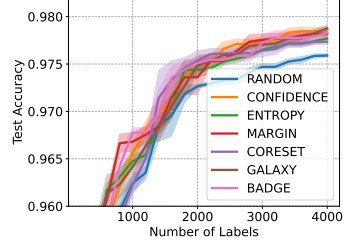 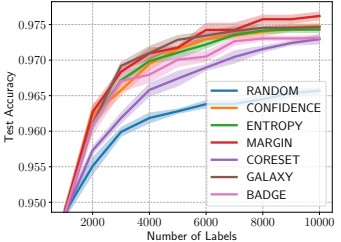

(a) Selection with shallow network, evaluation on fine-tuning, batch size of 1000

(b) Selection with shallow network, evaluation on fine-tuning, batch size of 200

(c) Generalization accuracy on CIFAR-10 with passive training.

Figure 5: (a) and (b): Generalization performance on CIFAR-10 when using different proxy models for data selection. (c): Generalization performance when using supervised trainer instead of Semi-SL (we use selection with end-to-end fine-tuning here). Each result is averaged over four trials with standard error shown as confidence interval.

### 4.4 Results and Discussion

In this section we present a summary of performance evaluations on various combinations of models and AL strategies.

**End-to-End Fine-Tuning.** First, we summarize our results when end-to-end fine-tuning the large pretrained model at every iteration of the data collection loop. When comparing the results of AL strategies to random sampling, we consistently see label efficiency gains across all datasets (Figures 1 and 4). Such label efficiency gain is especially significant on pool performances, with active learning strategies saving up to 50% of the annotation budget for ImageNet (Figure 4(d)). Notably, these gains are not confined to CLIP models. As shown in Figures 4(a,d), we also observe consistent gains in accuracies with the pretrained CoCa

| | Test Accuracy | | | Pool Accuracy | | |
|---|---|---|---|---|---|---|
| | Fine-tune | Shallow Network | Linear Probe | Fine-tune | Shallow Network | Linear Probe |
| Confidence | $\mathbf{77.38 \pm .13}$ | $76.96 \pm .12$ | $\mathbf{77.23 \pm .10}$ | $\mathbf{90.11 \pm .01}$ | $\mathbf{88.93 \pm .01}$ | $\mathbf{89.01 \pm .02}$ |
| Entropy | $77.12 \pm .04$ | $76.63 \pm .11$ | $76.81 \pm .01$ | $89.62 \pm .01$ | $88.33 \pm .02$ | $88.70 \pm .003$ |
| Margin | $77.37 \pm .04$ | $\mathbf{77.15 \pm .01}$ | $77.09 \pm .10$ | $90.02 \pm .03$ | $88.75 \pm .03$ | $88.84 \pm .01$ |
| Coreset | $75.54 \pm .15$ | $75.33 \pm .17$ | $75.54 \pm .08$ | $85.60 \pm .01$ | $84.84 \pm .03$ | $84.52 \pm .01$ |
| BADGE | $77.15 \pm .02$ | $76.83 \pm .04$ | $76.85 \pm .20$ | $89.10 \pm .04$ | $87.64 \pm .02$ | $88.20 \pm .03$ |
| Random | $76.12 \pm .14$ | $76.12 \pm .14$ | $76.12 \pm .14$ | $83.35 \pm .01$ | $83.35 \pm .01$ | $83.35 \pm .01$ |
| **Best** | $77.38 \pm .13$ | $77.15 \pm .01$ | $77.23 \pm .10$ | $90.11 \pm .01$ | $88.93 \pm .01$ | $89.01 \pm .02$ |

Table 2: Selection-via-proxy results of ImageNet using CLIP ViT-B32. The results are evaluated with 400,000 labels. Confidence intervals are standard errors based on two trials.

model. In general, when comparing performance of different AL strategies on (near) balanced datasets (ImageNet, CIFAR-10, CIFAR-100 and fMoW), margin sampling surprisingly performs among the top in terms of both generalization and pool accuracy. On imbalanced dataset like iWildcam (see Figure 7 in Appendix F), GALAXY demonstrates a clear advantage in terms of generalization and pool macro F1 scores. Finally, CORESET underperforms in most cases. These findings underscore the importance of further evaluating AL strategies on realistic datasets.

**Importance of AL + Semi-SL + Large Pretrained Models.** Comparing to existing literature of AL + Semi-SL (Lüth et al., 2023; Chan et al., 2021; Mittal et al., 2019; Siméoni et al., 2021) and AL + large pretrained models (Tamkin et al., 2022), our experiment yields the largest percentage of annotation cost savings to reach the same level of accuracy as random sampling. This reinforces the importance of studying the combination of active learning, semi-supervised learning and large pretrained models under an unified framework.

We compare the effect of using Semi-SL versus regular passive training when combined with AL + large pretrained models. By comparing Figure 1(a) with Figure 5(c), we see that the accuracy gains from each of AL and Semi-SL become less significant than the gains of each of them alone. However, the combination of both AL + Semi-SL provides the highest accuracy boost. Moreover, in terms of label savings in reaching the same accuracy, we find AL is the most efficient when combining with Semi-SL and large pretrained models, indicating the increasing importance of studying active learning in the new era of large pretrained models.

**Selection-via-proxy.** We also study the effectiveness and drawbacks of selection-via-proxy where we only retrain shallow neural networks or linear header (proxy models) for data selection. We compare it against *selection with end-to-end fine-tuning*, where one fine-tunes the entire model during the data collection process. Note that despite using different models for data selection, our evaluation results for both strategies are reported based on fine-tuning pretrained models end-to-end on the selected examples. As shown in Tables 2, 6, 9, selection-via-proxy performs similarly to selection with end-to-end fine-tuned models in terms of *test accuracy*. On the other hand, we found that selection-via-proxy is

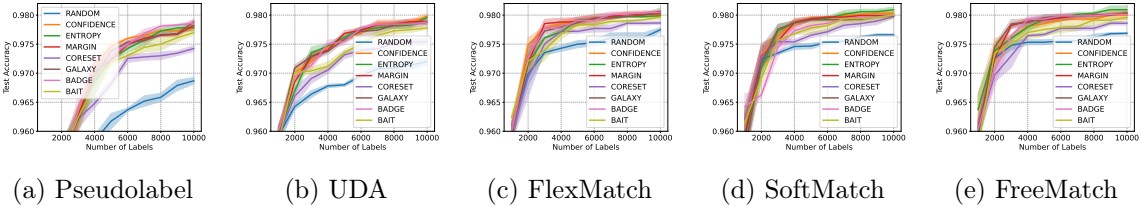

|                |                |                |                |                |
| :------------: | :------------: | :------------: | :------------: | :------------: |
| (a) Pseudolabel | (b) UDA | (c) FlexMatch | (d) SoftMatch | (e) FreeMatch |

Figure 6: Generalization Accuracy on CIFAR-10 with Alternate Semi-SL algorithms. Each result is averaged over three trials with standard error shown as the confidence interval.

slightly less effective than selection with fine-tuning in terms of *pool accuracy* - there is an approximately 1% reduction in performance in fMoW and ImageNet experiments.

To further investigate the label-efficiency tradeoff of the two methods, in Figure 1(a) and Figure 5(a), we plot their performances respectively after collecting every batch of labels. The gap between selection-via-proxy and selection with fine-tuning diminishes quickly with more iterations of data selection. As shown in Figure 5(b), we can further close the gap in lower-budget settings by collecting more rounds of annotations with smaller batches. Indeed, to achieve 97.75% accuracy (random sampling's accuracy with 10,000 labels), selection-via-proxy only requires 2750 labels (with batch size of 200), comparable to selection with fine-tuning's label-efficiency in Figure 5(a). We note that smaller batches are only computationally feasible for selection-via-proxy, as one can only end-to-end fine-tune a small number of times under a limited budget.

**Importance of the Choice of Semi-SL** We evaluate the effect of using alternative Semi-SL techniques for end-to-end finetuning on CIFAR-10. We compare FlexMatch against two common Semi-SL baselines, UDA (Xie et al., 2020a) and Pseudolabeling (Lee, 2013), and two recent Semi=SL approaches, SoftMatch (Chen et al., 2023) and FreeMatch (Wang et al., 2023), on CIFAR-10 in Figure 6. FlexMatch, FreeMatch, and SoftMatch achieve higher accuracy than UDA and Pseudolabeling across all AL sampling strategies. On CIFAR-10, the differences are most pronounced in the early AL rounds but diminish during the later rounds. Notably, label efficiency seems to be affected more by the choice of Semi-SL method than by the choice of AL method. Similar to the generalization accuracies in Figure 6, we report pool accuracies in Figure 21.

Additionally, the results also highlight a consistent trend in the relative performance of AL methods across the Semi-SL techniques. Regardless of the specific Semi-SL method employed, we consistently observe that AL methods outperforms random sampling. Moreover, the AL strategies demonstrate a level of transferability across different Semi-SL methods. Namely, AL strategies tend to perform similar relative to each other regardless of the Semi-SL algorithm used during training. Similarly, the relative performance of Semi-SL algorithms stays the same while varying different active learning strategies. This suggests research in the two respective fields can be conducted separately while incorporating only the state-of-the-art method from the other field under the LabelBench framework.

**More Results.** See Appendix F for additional results that support both the above and more findings.

## 5 Call for Contribution and Future Work

We call on the broader community to further develop different components of LabelBench: below we provide suggested contributions for each directory of our framework. Our codebase is modular, so one can easily start on any single component without a thorough understanding of the entire codebase.

**Trainer.** Our experiments demonstrate the potential label savings provided by combining active learning with FreeMatch, SoftMatch, FlexMatch, UDA, and Pseudolabeling. To expand upon these results, it would be valuable to develop a benchmark of additional Semi-SL methods, evaluated in combination with AL and large pretrained models. To do so, one could instantiate specific Semi-SL trainer classes that inherit our template Semi-SL trainer.

**Active Learning Strategy.** As exhibited by our results, combining AL with Semi-SL and large pretrained models results in highly accurate and label-efficient models that demonstrate clear gains over passive learning. Therefore, we call on the AL community to evaluate their active selection algorithms under our more comprehensive and up-to-date benchmark. Additionally, besides several widely-used AL strategies we have implemented here, there exists a large suite of active selection methods in the literature that could potentially be implemented and evaluated (Ren et al., 2021).

**Datasets and Metrics.** In future benchmarks built on top of LabelBench, we plan on incorporating datasets with distribution shift evaluation data. We believe this is a valuable future direction that aligns with many real-world scenarios. Introducing tasks beyond image classification is also an important next step, e.g., natural language processing tasks, vision tasks such as object detection and segmentation, and generative modeling in both vision and language applications.

**Models.** With the computational speed-ups afforded by selection-via-proxy and our pre-computation steps, we can scale the model to ViT-L and ViT-H architectures (Dosovitskiy et al., 2020) without incurring significant computational costs. In addition, there are also other choices of proxy. For example, LORA (Hu et al., 2021), which injects the trainable rank decomposition matrices in the intermediate layers instead of the final header is popular in natural language processing. Moreover, in developing benchmarks that better reflect real-world applications, contributors could implement a blend of selection-via-proxy and end-to-end fine-tuning during example selection, instead of fine-tuning at every iteration or only once at the end of labeling.

**Weak Supervision.** Utilizing weak annotation sources to annotate is another growing subfield of label-efficient learning (Zhang et al., 2021b). Incorporating weak supervision with large pretrained models, Semi-SL and AL poses many open questions. We believe incorporating weak supervision in a rigorous way is an important next step in improving LabelBench.

## 6 Conclusion

In this paper, we present LabelBench, a comprehensive and computationally efficient framework for evaluating label-efficient learning. A key challenge we address in this paper is the computational complexity of retraining large pretrained models when using active learning. We find selection-via-proxy with probes provides strong generalization performances on a

variety of datasets. We believe retraining with both end-to-end fine-tuning and selection-via-proxy are important settings. In practice, one may make the decision of whether to use a proxy model based on the tradeoff between model training cost and annotation cost. We encourage the active learning research community to further investigate and propose algorithms under the selection-via-proxy setting and utilize our light-weight benchmarks.

LabelBench puts label-efficient learning under the spotlight of fine-tuning large pretrained model. A pivotal realization from our experiments is the necessity to re-calibrate our focus. Beyond algorithm development in isolated research areas, it is crucial to study how existing tools – such as pretrained models, semi-supervised learning, and active learning – can be skillfully intertwined and leveraged together.

## Broader Impact Statement

Research into Label-Efficient learning methods to reduce annotation cost of large pretrained models has far-reaching implications.

Benefits of Label-Efficient learning include:

1. Resource Efficiency: Label-efficient learning can significantly reduce computational and data resources. This can lower the barriers to entry for smaller organizations and researchers, democratizing access to advanced AI technologies.
2. Environmental Impact: Less computational power means a reduced carbon footprint associated with training large models, which is crucial given the increasing concerns about the environmental impact of AI.
3. Data Quality over Quantity: This approach emphasizes the importance of high-quality, informative data over sheer volume. It can lead to more effective and efficient learning processes.
4. Rapid Adaptation and Deployment: Organizations can quickly adapt models to new tasks or domains with fewer examples, accelerating the pace of innovation and deployment of AI solutions.
5. Enhanced Personalization: With the need for fewer data, it becomes easier to fine-tune models for specific, niche applications, enhancing personalization and user experience.

Risks of Label-Efficient Learning include:

1. Bias and Representativeness: If the subset of data used for training is not representative, there's a risk of amplifying biases. Care must be taken to ensure the selected training examples are diverse and inclusive.
2. Overfitting: With fewer examples, there's a risk that the model may overfit to the training data, leading to poor generalization on unseen data.
3. Security and Privacy: Focusing on a smaller set of data could potentially make it easier to reverse-engineer sensitive information, raising privacy concerns.
4. Dependence on Initial Data Quality: The effectiveness of label-efficient learning heavily depends on the initial quality of examples. Poor initial data quality could lead to suboptimal learning.
5. Complexity in Implementation: Label-efficient learning strategies can be complex to implement and might require domain-specific knowledge to identify the most informative data points.

## Acknowledgments and Disclosure of Funding

This work is supported by AFOSR/AFRL grant FA9550-18-1-0166 and by the NSF under Grant Nos. CNS-2112471, IIS-2106937 and IIS-2148367.

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

## Appendix A. Definition of Metrics

For $K$ labels, define the confusion matrix $C$ where $C_{i,j} = \Pr(Y = i, \hat{Y} = j)$.

The balanced accuracy is

$$\frac{1}{K} \sum_{i=1}^{K} \frac{C_{i,i}}{\sum_{j=1}^{K} C_{i,j}} \tag{1}$$

Define the precision and recall for a class $i$ as

$$P_i = \frac{C_{i,i}}{\sum_{j=1}^{K} C_{j,i}} \tag{2}$$

$$R_i = \frac{C_{i,i}}{\sum_{j=1}^{K} C_{i,j}} \tag{3}$$

Then, the macro F1 score is

$$\frac{1}{K} \sum_{i=1}^{K} \frac{2}{\frac{1}{P_i} + \frac{1}{R_i}} \tag{4}$$

## Appendix B. Active Learning Strategies

We describe the active learning setup and introduce some basic active learning strategies in this section.

We start by describing the active learning setups. The learner starts with a large pool of unlabeled examples $U = \{x_i\}_{i \in [n]}$ and a small fraction of labeled examples $L$, where each example $x$ comes from the input space $\mathcal{X}$ with some unknown label $y$ belonging to labeling space $\mathcal{Y}$. At the beginning of every batch, adhering to a certain active learning strategy, the algorithm adaptively selects new examples to label (i.e., moving the labeled examples from $U$ to $L$) based on the current model $h$. We use $h_\theta(x)$ to denote the predicated softmax vector; we also use $[h_\theta(x)]_i$ to denote the $i$-th coordinate of the prediction. The model $h$ is then retrained based on the updated dataset $L, U$ with a certain training strategy. The ultimate goal is to use as small of a labeling budget as possible to achieve some desired performance (e.g., small error).

Below we introduce some active learning strategies that have been used in our experiments.

- Confidence (Lewis, 1995): An uncertainty-based active learning strategy that selects examples with the least confidence score in terms of the top predicated class, i.e., $\max_i [h_\theta(x)]_i$.
- Entropy (Settles, 2009): An uncertainty-based active learning strategy that selects examples with the highest entropy of the predicted distribution $h_\theta(x)$.
- Margin (Scheffer et al., 2001): An uncertainty-based active learning strategy that selects examples with the smallest prediction margin between the top-2 classes, i.e., $[h_\theta(x)]_{i^\star} - \max_{i \neq i^\star} [h_\theta(x)]_i$, where $i^\star = \arg\max[h_\theta(x)]_i$.
- CORESET (Sener and Savarese, 2017): A diversity-based active learning strategy that selects samples by approximating the solution to a k-Centers objective function.

- BADGE (Ash et al., 2019): An active learning strategy that incorporates both uncertainty and diversity in sampling using k-means++ in the hallucinated gradient space.
- BAIT (Ash et al., 2021): An active learning strategy that incorporates both uncertainty and diversity by sampling from a Fisher-based selection objective using experimental design. BAIT can be viewed as a more general version of BADGE.
- GALAXY (Zhang et al., 2022): A graph-based active learning strategy that incorporates both uncertainty and diversity by first building a graph and then adaptively sampling examples on the shortest path of the graph.

## Appendix C. Semi-Supervised Learning Strategies

Semi-SL methods are used when there is a large unlabeled pool $U$ and a small labeled pool $L$. Semi-SL methods all apply some form of supervised loss to the labeled samples, and typically differ in how they utilize unlabeled samples. Below, we provide a brief description of the Semi-SL methods that we considered:

- Pseudolabeling (Lee, 2013): A pseudolabeling based semi-supervised learning method that assigns pseudolabels to unlabeled samples on which model confidence exceeds a fixed threshold.
- UDA (Xie et al., 2020b): A consistency-regularization based semi-supervised learning method that ensures that the model predictions are consistent on both weakly and strongly augmented versions of highly confident unlabeled samples.
- FlexMatch (Zhang et al., 2021a): A semi-supervised learning method that uses both consistency-regularization (similar to UDA) and pseudolabeling on unlabeled samples. Unlike UDA and Pseudolabeling, this approach also uses a dynamic confidence threshold, dependent on both time and class, to select which unlabeled samples to use in the unsupervised loss.
- FreeMatch (Wang et al., 2022c): Like FlexMatch, FreeMatch is a semi-supervised learning method that uses both consistency-regularization (similar to UDA) and pseudolabeling on unlabeled samples. In addition to an adaptive threshold for assigning pseudo labels for every class, FreeMatch also utilize class-specific adaptive thresholds to encourse class diversity.
- SoftMatch (Chen et al., 2023): While SoftMatch uses a consistency-regularization regime as UDA, FlexMatch and FreeMatch above, it uses a soft threshold technique in generating pseudo labels. Specifically, the small amount of high confidence examples are weighted higher while the vast amount of lower confidence examples are also "pseudo-labeled", but weighted less aggressively.

## Appendix D. Hyper-parameter tuning

Adhering to the guidelines proposed by Lüth et al. (2023), we are transparent about our method configuration, which many active learning studies fail to report. For each dataset, we utilize a separate validation set, typically with size around 10% of the training pool. We begin the process by adjusting the hyper-parameters on a subset of the training data, which is randomly queried and constitutes around 10% of the total training pool. The selection of hyper-parameters is mainly based on the criterion of achieving the highest accuracy on the

validation set. These hyper-parameters are then consistently applied in all subsequent data collection batches and across varied experimental settings (e.g., experiments with different batch sizes). While it's arguable that this fixed hyper-parameter approach may not always yield optimal results, it is practically suitable in real-world scenarios and allows for fair comparison in this paper.

## Appendix E. Speeding Up Existing Active Learning Algorithms

**Notation.** Let $U = \{x_1, ..., x_N\}$ denote the set of $N$ unlabeled examples and $K$ denote the number of classes in a dataset. For each $i \in [N]$, we further use $p_i \in \mathbb{R}^K$ and $\widehat{y}_i \in [K]$ to denote the predictive probability and predictive label respectively on example $x_i$. Lastly, we use $v_1, ..., v_N \in \mathbb{R}^d$ to denote the penultimate layer output of a neural network where $d$ is the number of dimensions.

**Implementation of BADGE.** The current implementation of BADGE (`https://github.com/JordanAsh/badge`) explicitly computes gradient embeddings $g_i$ for each unlabeled example $x_i$. In particular, each $g_i$ is a $Kd$-dimensional vector and can be computed via vectorizing $q_i v_i^\top$ where $q_i \in \mathbb{R}^K$ is defined as

$$q_{i,j} = \begin{cases} 1 - p_{i,j} & \text{if} \quad j = \widehat{y}_i \\ -p_{i,j} & \text{otherwise} \end{cases}$$

During each iteration of BADGE ($B$ iterations in total for each batched selection of $B$ examples), the dominating computation lies in computing the $\ell$-2 distance between $N$ pairs of gradient embeddings. Currently, this is implemented by naively computing $\|g_i - g_j\|_2$ with an $O(Kd)$ complexity each.

We instead use the following decomposition:

$$\begin{aligned} \|g_i - g_j\|_2 &= \|g_i\|_2 + \|g_j\|_2 - 2g_i^\top g_j \\ &= \|q_i\|_2 \cdot \|v_i\|_2 + \|q_j\|_2 \cdot \|v_j\|_2 - 2 \cdot (q_i^\top q_j) \cdot (v_i^\top v_j). \end{aligned}$$

where the last expression can be computed with $O(K + d)$ complexity, effectively reducing the computational time by an order of magnitude. In our ImageNet experiment, this means a 512-fold reduction in computation time.

**Implementation of BAIT.** The current implementation of BAIT (`https://github.com/JordanAsh/badge`) uses an apparent approximation to the Fisher information for a low-rank approximation. Note that the multi-class Fisher information defined in appendix A.2 of Ash et al. (2021) is not full-rank, causing numerical problems with taking the inverse. In our implementation, we multiply the Fisher information by a orthogonal transformation that removes a dimension to make the Fisher information full-rank.

Define the orthogonal transformation as $T \in \mathbb{R}^{k \times (k-1)}$ that removes the null space along the direction of the vectors of all ones. Using the notation of appendix A.2 of Ash et al. (2021), we can let:

$$P = T^\top (\text{diag}(\pi) - \pi\pi^\top) T \tag{5}$$

$$U = x \otimes P^{1/2} \tag{6}$$

Then,

$$UU^\top = (x \otimes P^{1/2})(x \otimes P^{1/2})^\top \qquad (7)$$
$$= xx^\top \otimes P \qquad (8)$$
$$= I(x; W) \qquad (9)$$

and thus we can use the Woodbury matrix identity for faster matrix inverse updates.

Because the Fisher information matrix is very large, we perform PCA to reduce the dimensionality.

In Ash et al. (2021), an expensive greedy strategy is used to build the selected set. Our implementation is based on "swaps", that is, removing an example and adding an example. In particular, we begin with an initially randomly drawn selected set, then one-by-one propose an example to remove and propose to add the best example from a random sample of 10 unlabeled examples. If the proposed swap would improve the objective function, the swap is performed.

## Appendix F. More results

Here we provide more experimental results. Notice that we only implement BAIT in CIFAR-10 due to its high computational and memory complexity – For $d$ embedding dimension and $K$-classes, its memory complexity is $\mathcal{O}(K^2 d^2)$. In addition, we omit GALAXY for ImageNet as mentioned in the main paper due to its expensive computational complexity on large datasets.

We also note that results on iWildcam have much higher standard error and variance than other datasets. We attribute this observation to the imbalance nature of the dataset, which may increase the variance if some rare classes have no annotated examples at all.

### F.1 End-to-end Fine-tuning

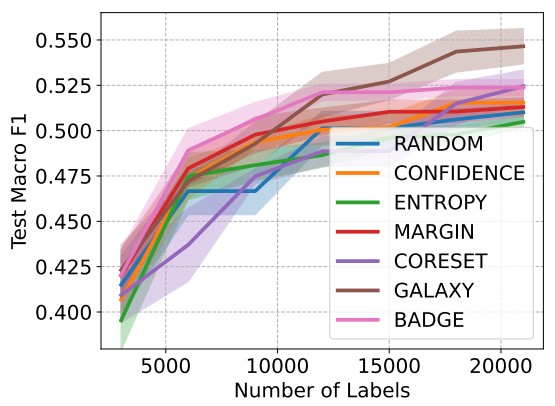

(a) Generalization macro F1 on iWildcam, AL + FlexMatch + Pretrained CLIP ViT-B32

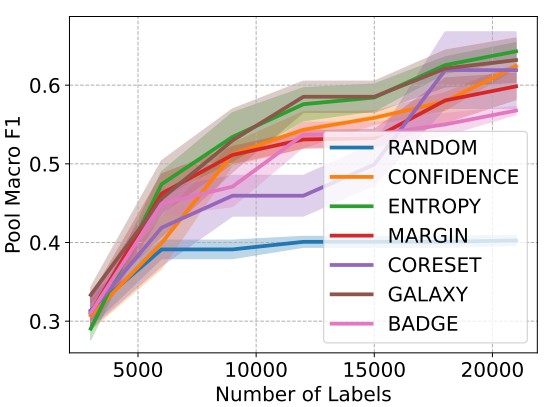

(b) Pool macro F1 on iWildcam

Figure 7: End-to-end fine-tune performance on iWildcam, AL + FlexMatch + Pretrained CLIP ViT-B32

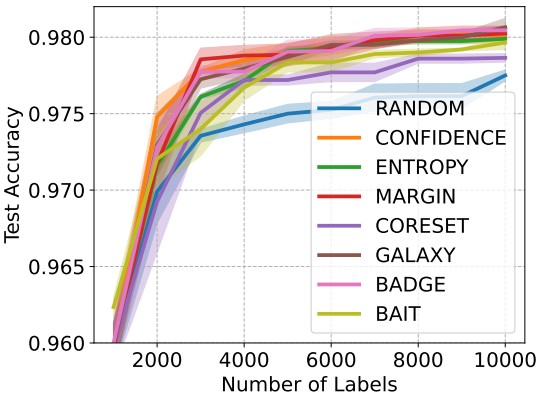

(a) Generalization Accuracy on CIFAR-10, AL + FlexMatch + Pretrained CLIP ViT-B32, Batch Size = 1000

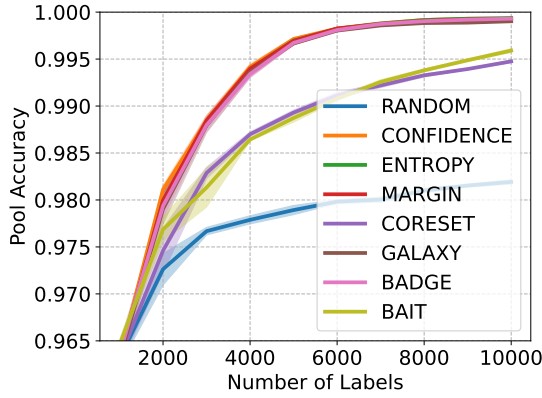

(b) Pool Accuracy on CIFAR-10, AL + FlexMatch + Pretrained CLIP ViT-B32, Batch Size = 1000

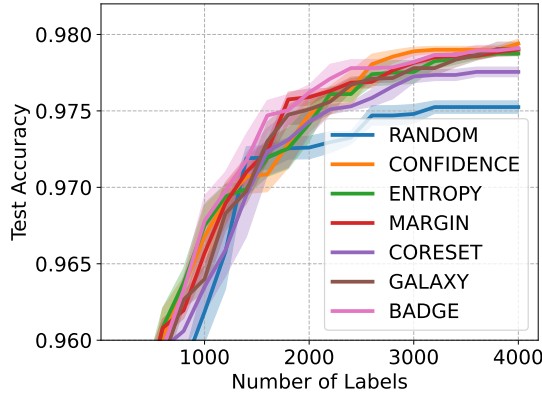

(c) Generalization Accuracy on CIFAR-10, AL + FlexMatch + Pretrained CLIP ViT-B32, Batch Size = 200

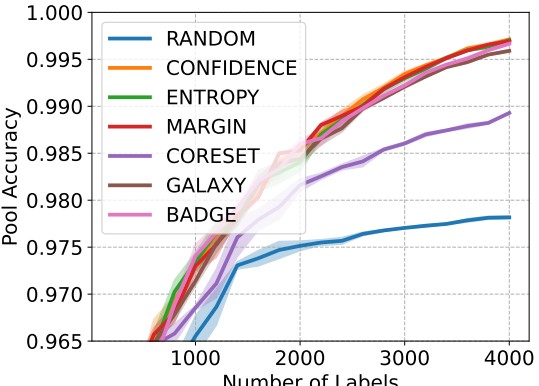

(d) Pool Accuracy on CIFAR-10, AL + FlexMatch + Pretrained CLIP ViT-B32, Batch Size = 200

Figure 8: End-to-end fine-tune performance on CIFAR-10.

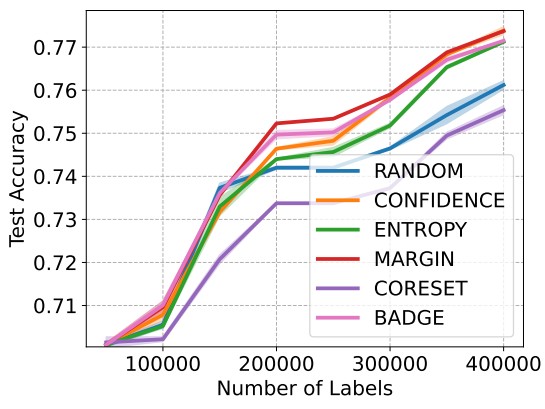

(a) Generalization Accuracy on ImageNet, AL + FlexMatch + Pretrained CLIP ViT-B32

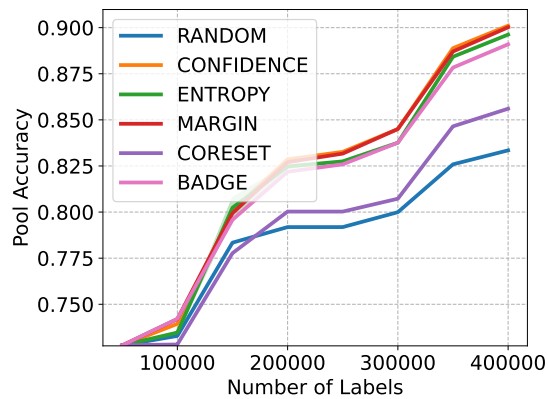

(b) Pool Accuracy on ImageNet, AL + FlexMatch + Pretrained CLIP ViT-B32

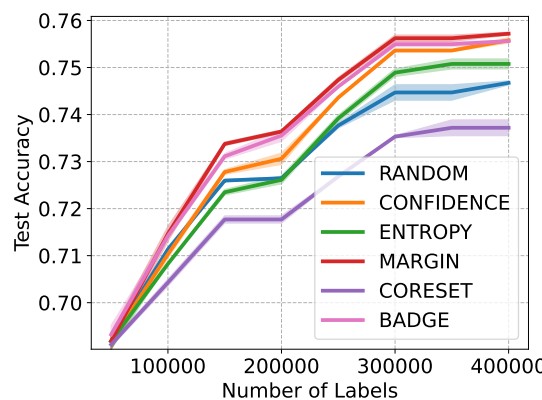

(c) Generalization Accuracy on ImageNet, AL + FlexMatch + Pretrained CoCa ViT-B32

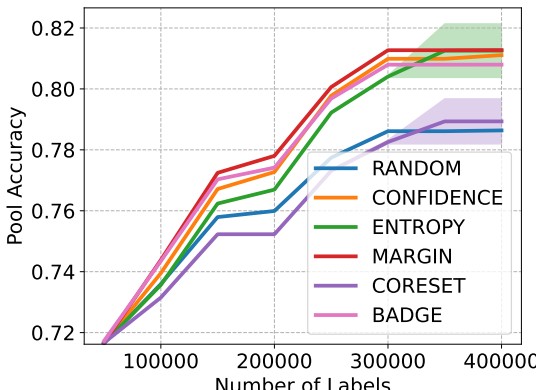

(d) Pool Accuracy on ImageNet, AL + FlexMatch + Pretrained CoCa ViT-B32

Figure 9: End-to-end fine-tune performance on ImageNet.

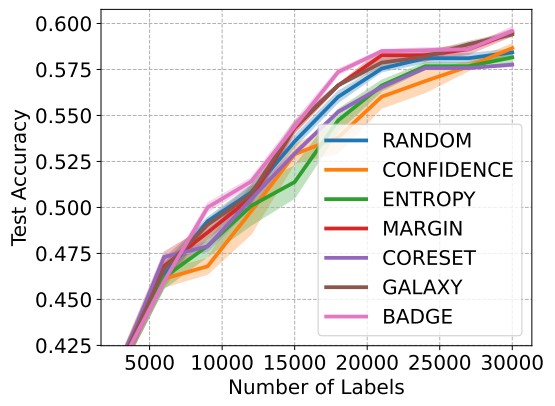 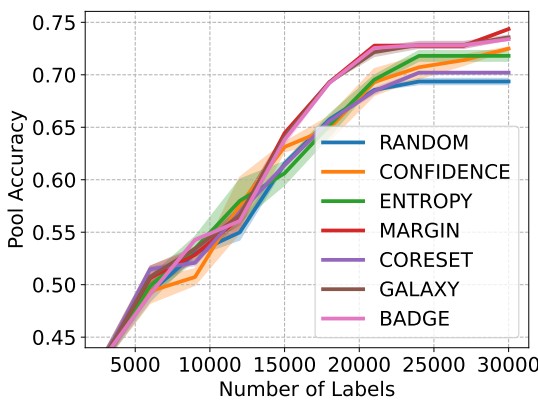

(a) Generalization Accuracy on FMoW, AL + Flex-Match + Pretrained CLIP ViT-B32

(b) Pool Accuracy on FMoW, AL + FlexMatch + Pretrained CLIP ViT-B32

Figure 10: End-to-end fine-tune performance on FMoW.

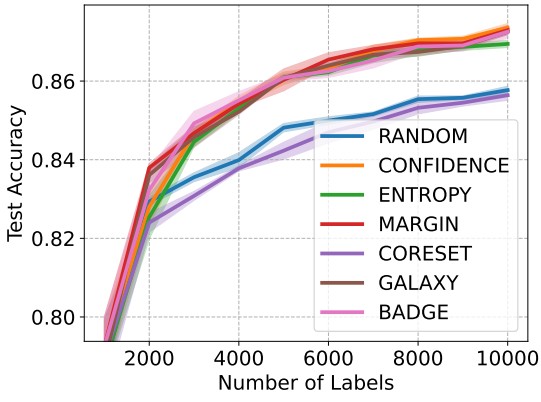 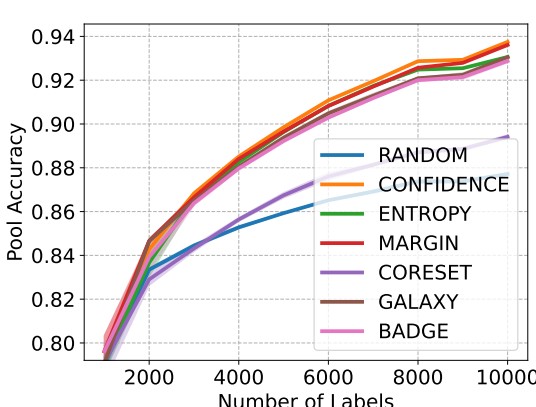

(a) Generalization Accuracy on CIFAR-100, AL + FlexMatch + Pretrained CLIP ViT-B32

(b) Pool Accuracy on CIFAR-100, AL + FlexMatch + Pretrained CLIP ViT-B32

Figure 11: End-to-end fine-tune performance on CIFAR-100.

## F.2 Learning Linear Probes

Note this section differs from the selection-via-proxy plots (Figures 5(a,b)) in that we are measuring the raw performance of linear probes instead of having an additional evaluation step by fine-tuning the model end-to-end.

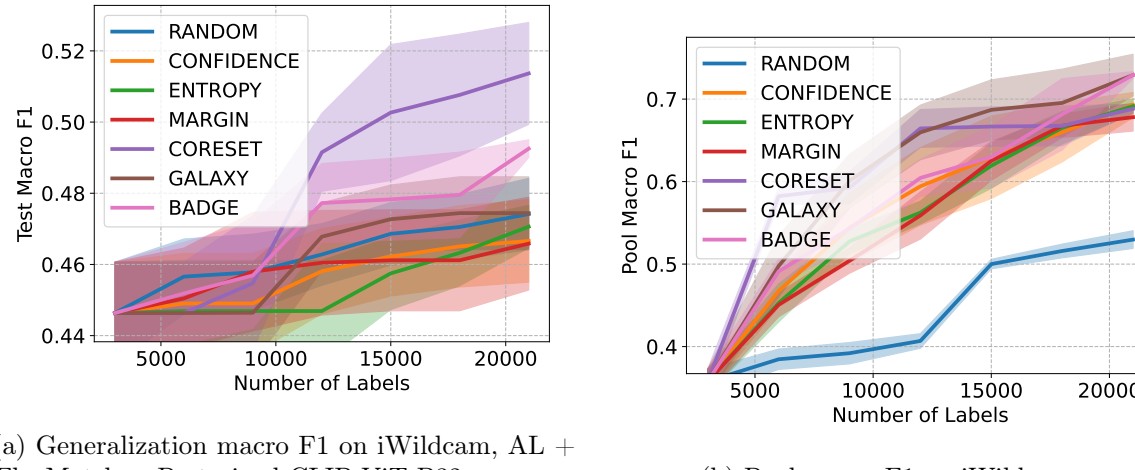

(a) Generalization macro F1 on iWildcam, AL + FlexMatch + Pretrained CLIP ViT-B32

(b) Pool macro F1 on iWildcam

Figure 12: Linear probe performance on iWildcam, AL + FlexMatch + Pretrained CLIP ViT-B32

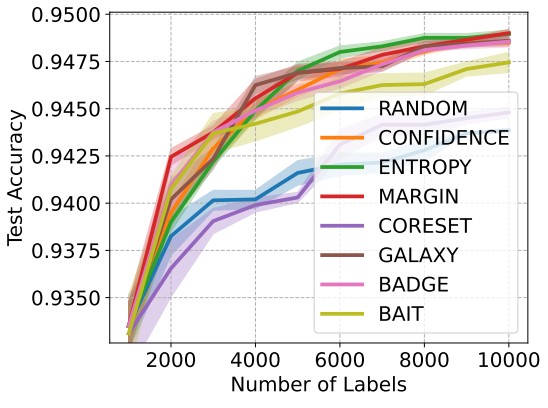

(a) Generalization Accuracy on CIFAR-10, AL + FlexMatch + Pretrained CLIP ViT-B32, Batch Size = 1000

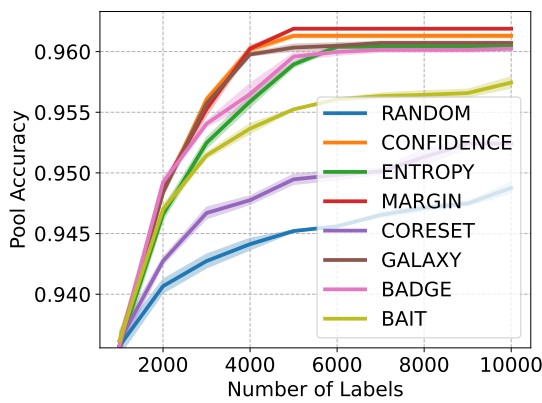

(b) Pool Accuracy on CIFAR-10, AL + FlexMatch + Pretrained CLIP ViT-B32, Batch Size = 1000

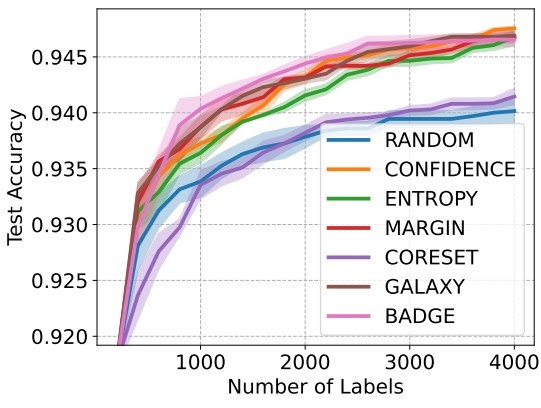

(c) Generalization Accuracy on CIFAR-10, AL + FlexMatch + Pretrained CLIP ViT-B32, Batch Size = 200

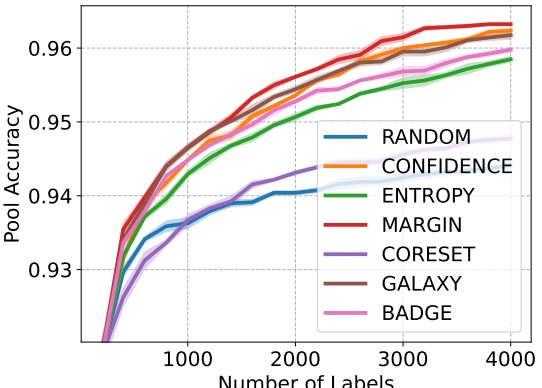

(d) Pool Accuracy on CIFAR-10, AL + FlexMatch + Pretrained CLIP ViT-B32, Batch Size = 200

Figure 13: Linear probe performance on CIFAR-10.

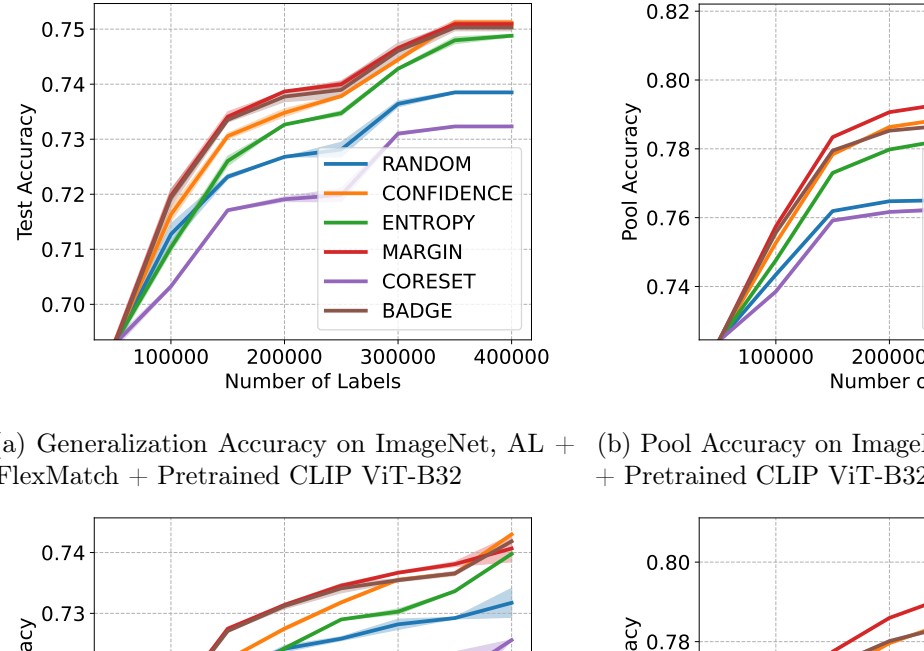

(a) Generalization Accuracy on ImageNet, AL + FlexMatch + Pretrained CLIP ViT-B32

(b) Pool Accuracy on ImageNet, AL + FlexMatch + Pretrained CLIP ViT-B32

(c) Generalization Accuracy on ImageNet, AL + FlexMatch + Pretrained CoCa ViT-B32

(d) Pool Accuracy on ImageNet, AL + FlexMatch + Pretrained CoCa ViT-B32

Figure 14: Linear probe performance on ImageNet.

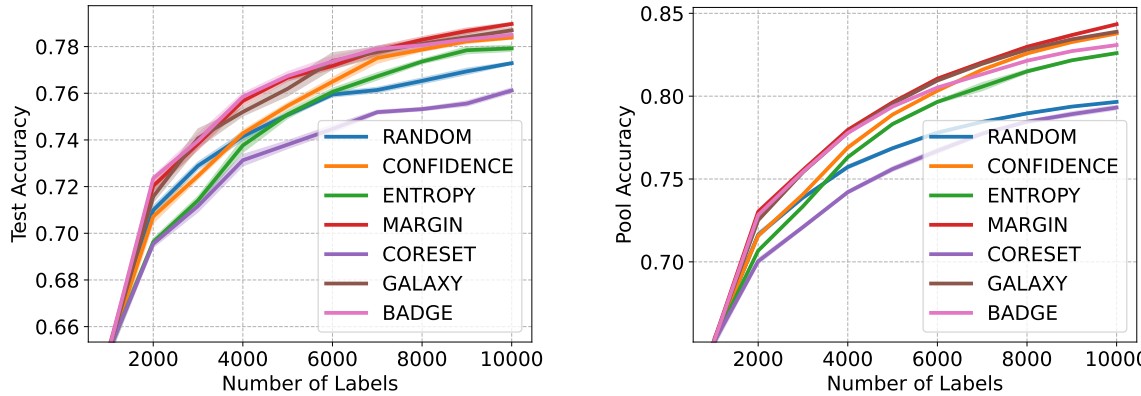

(a) Generalization Accuracy on CIFAR-100, AL + FlexMatch + Pretrained CLIP ViT-B32

(b) Pool Accuracy on CIFAR-100, AL + FlexMatch + Pretrained CLIP ViT-B32

Figure 15: Linear probe performance on CIFAR-100.

### F.3 Learning a Shallow Neural Network

Note this section differs from the selection-via-proxy plots (Figures 5(a,b)) in that we are measuring the raw performance of shallow networks instead of having an additional evaluation step by fine-tuning the model end-to-end.

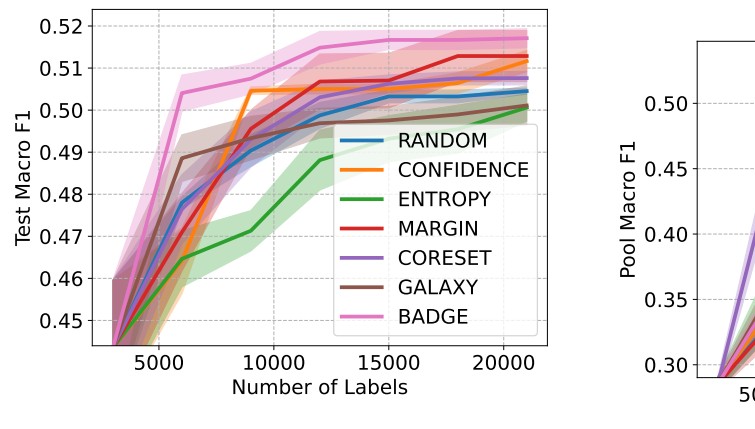
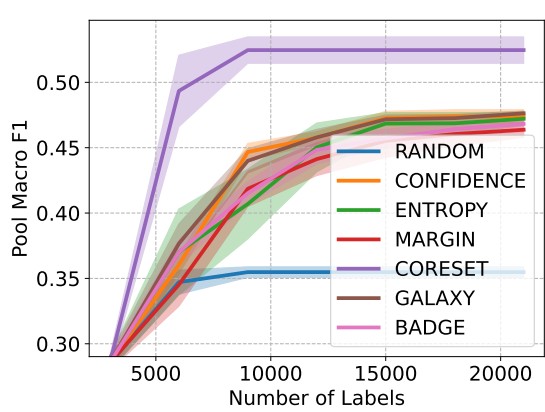

(a) Generalization macro F1 on iWildcam, AL + FlexMatch + Pretrained CLIP ViT-B32

(b) Pool macro F1 on iWildcam

Figure 16: Shallow network performance on iWildcam, AL + FlexMatch + Pretrained CLIP ViT-B32

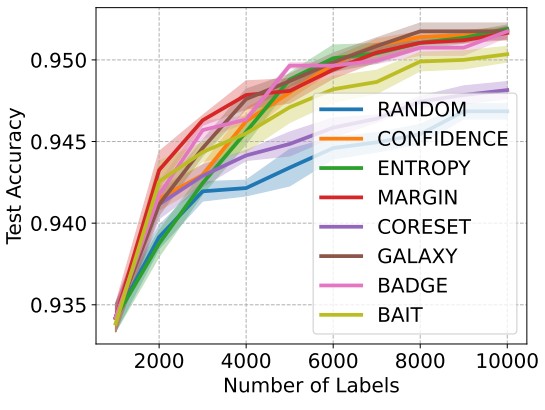

(a) Generalization Accuracy on CIFAR-10, AL + FlexMatch + Pretrained CLIP ViT-B32, Batch Size = 1000

(b) Pool Accuracy on CIFAR-10, AL + FlexMatch + Pretrained CLIP ViT-B32, Batch Size = 1000

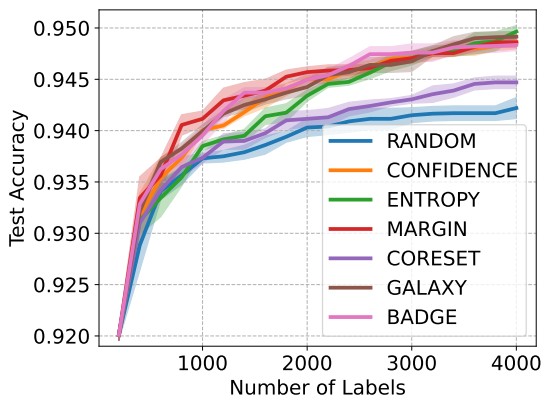

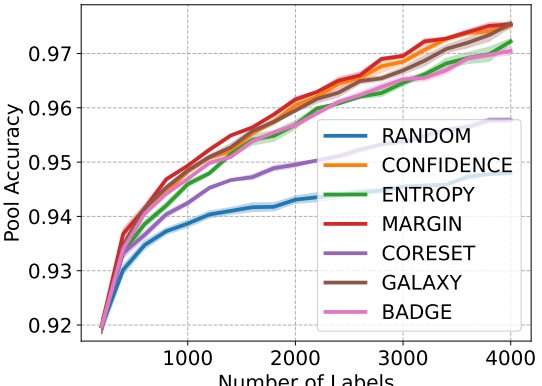

(c) Generalization Accuracy on CIFAR-10, AL + FlexMatch + Pretrained CLIP ViT-B32, Batch Size = 200

(d) Pool Accuracy on CIFAR-10, AL + FlexMatch + Pretrained CLIP ViT-B32, Batch Size = 200

Figure 17: Shallow network performance on CIFAR-10.

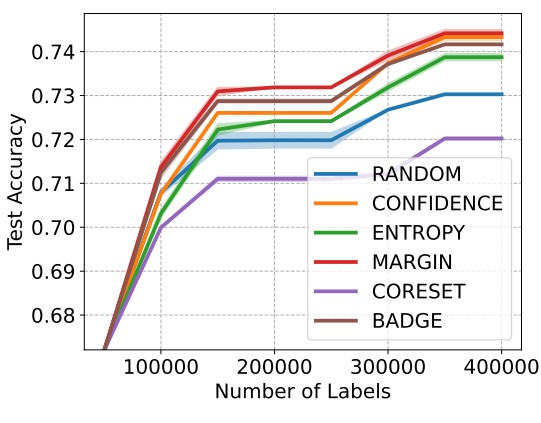

(a) Generalization Accuracy on ImageNet, AL + FlexMatch + Pretrained CLIP ViT-B32

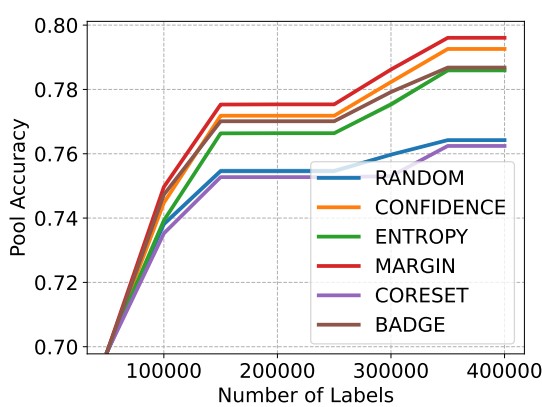

(b) Pool Accuracy on ImageNet, AL + FlexMatch + Pretrained CLIP ViT-B32

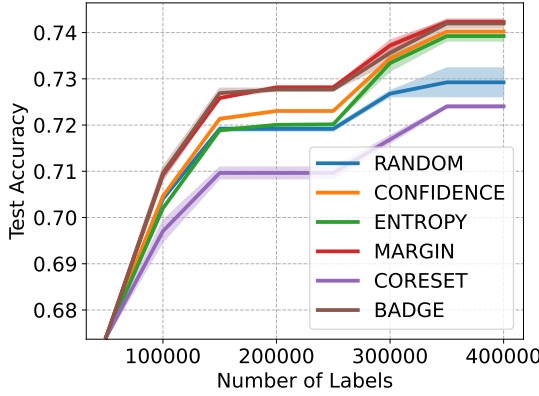

(c) Generalization Accuracy on ImageNet, AL + FlexMatch + Pretrained CoCa ViT-B32

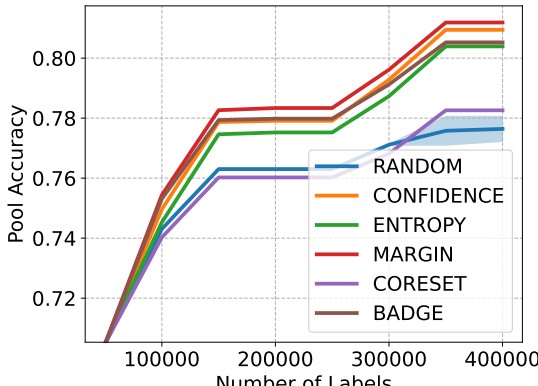

(d) Pool Accuracy on ImageNet, AL + FlexMatch + Pretrained CoCa ViT-B32

Figure 18: Shallow network performance on ImageNet.

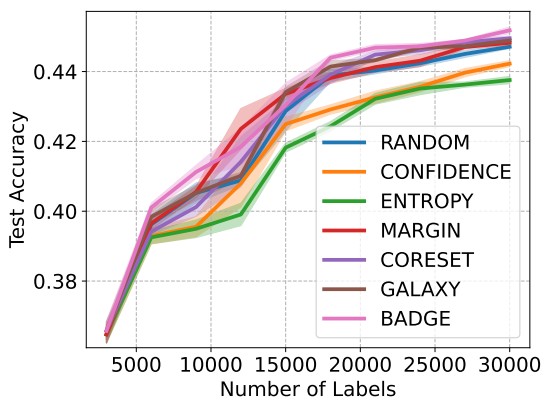
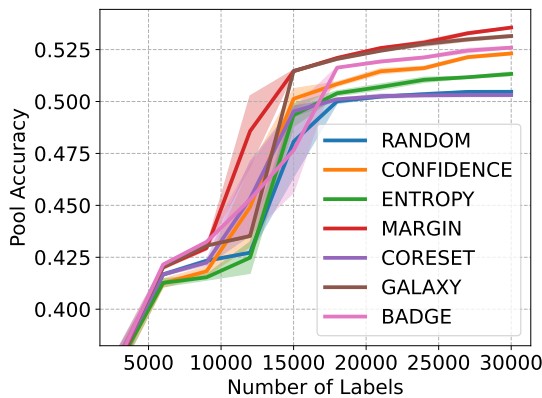

(a) Generalization Accuracy on FMoW, AL + Flex-Match + Pretrained CLIP ViT-B32

(b) Pool Accuracy on FMoW, AL + FlexMatch + Pretrained CLIP ViT-B32

Figure 19: Shallow network performance on FMoW.

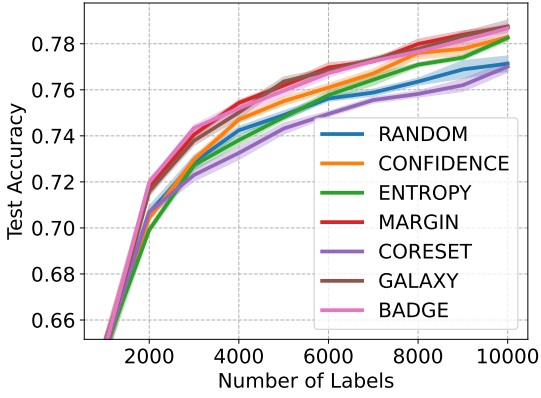
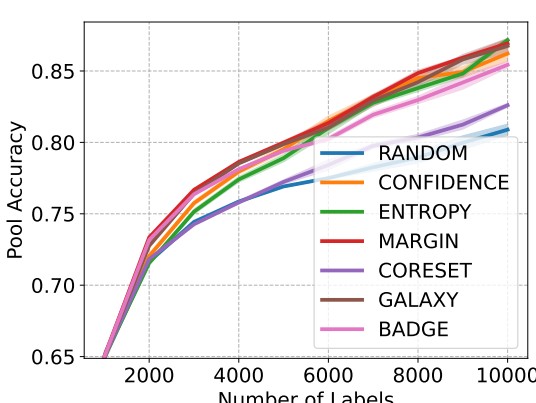

(a) Generalization Accuracy on CIFAR-100, AL + FlexMatch + Pretrained CLIP ViT-B32

(b) Pool Accuracy on CIFAR-100, AL + FlexMatch + Pretrained CLIP ViT-B32

Figure 20: Shallow network performance on CIFAR-100.

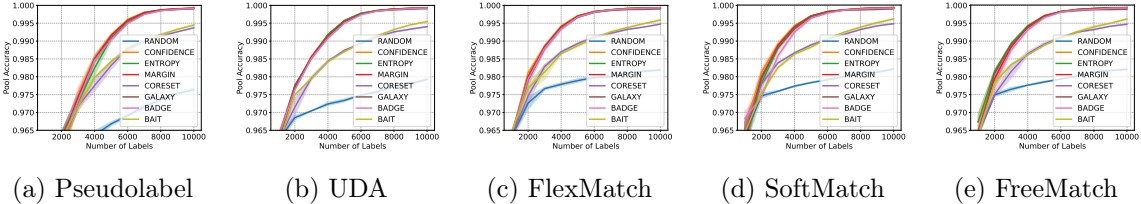

Figure 21: Pool Accuracy on CIFAR-10 with Alternate Semi-SL algorithms. Each result is averaged over three trials with standard error shown as the confidence interval.

## F.4 Additional Results for End-to-end Fine-tuning with different Semi-SL Methods

Here we evaluate the effect of using alternative Semi-SL techniques on the pool accuracy for the end-to-end finetuning on CIFAR10 in Figure 21. Furthermore, we include results for CIFAR100 on with Pseudolabeling, UDA, and FlexMatch in Figure 22.

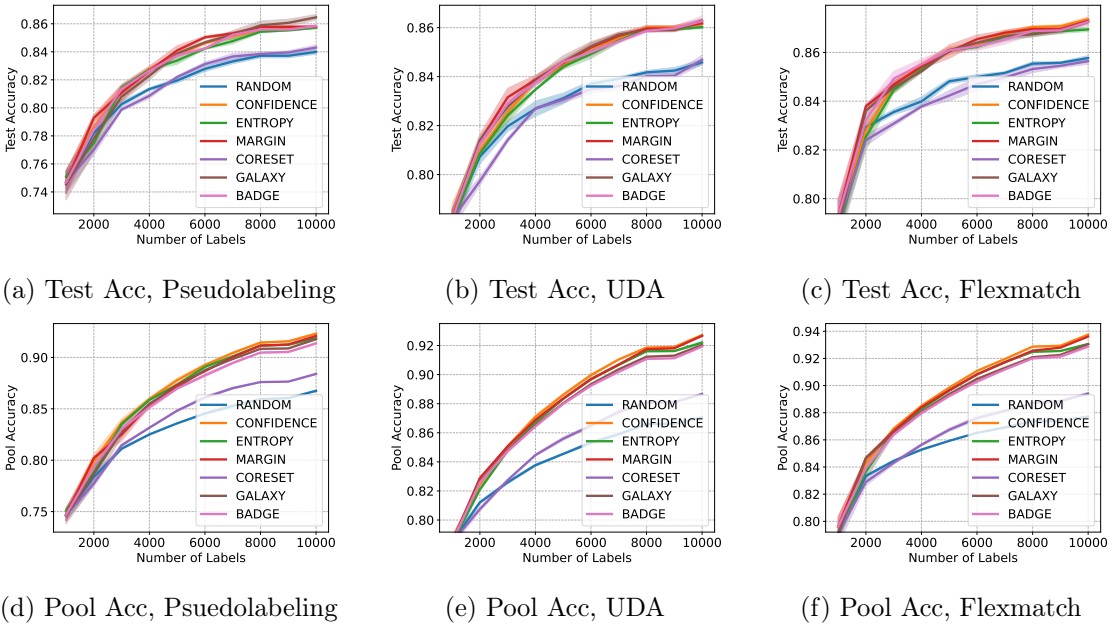

Figure 22: Results on CIFAR100 with different Semi-SL algorithms. Similar to the results of CIFAR10, we find that the choice of Semi-SL algorithm is very salient.

## F.5 Evaluation on Small Budgets

The standard AL setup in recent works (Beck et al., 2021; Coleman et al., 2019) uses significantly larger labeling budgets than the standard Semi-SL setup (Sohn et al., 2020; Zhang et al., 2021a; Lee, 2013; Xie et al., 2020b). In Table 3 and 4, we experiment with AL methods in the small budget setting and demonstrate that AL still demonstrates considerable gains in accuracy compared to random sampling.

| | Test Accuracy | | | Pool Accuracy | | |
|---|---|---|---|---|---|---|
| | Pseudolabeling | UDA | FlexMatch | Pseudolabeling | UDA | FlexMatch |
| Confidence | $88.78 \pm .84$ | $88.84 \pm 1.0$ | $\mathbf{91.51 \pm .93}$ | $89.28 \pm .81$ | $88.75 \pm 1.1$ | $\mathbf{91.70 \pm .80}$ |
| Entropy | $89.91 \pm 1.5$ | $\mathbf{90.78 \pm .67}$ | $90.90 \pm 2.6$ | $90.14 \pm 1.5$ | $\mathbf{90.91 \pm .62}$ | $91.53 \pm 2.4$ |
| Margin | $89.07 \pm .03$ | $88.36 \pm 2.1$ | $90.73 \pm 1.2$ | $89.23 \pm .24$ | $88.58 \pm .08$ | $90.79 \pm 1.4$ |
| Coreset | $88.66 \pm 3.8$ | $88.86 \pm 3.9$ | $91.03 \pm 1.5$ | $88.70 \pm 4.0$ | $89.04 \pm 4.3$ | $91.28 \pm 1.6$ |
| BADGE | $\mathbf{90.56 \pm .11}$ | $90.01 \pm .44$ | $91.41 \pm .23$ | $\mathbf{90.84 \pm .24}$ | $90.70 \pm .20$ | $91.56 \pm .34$ |
| Random | $83.25 \pm 3.2$ | $83.21 \pm 3.3$ | $90.9 \pm 2.6$ | $84.27 \pm 2.81$ | $84.24 \pm 2.8$ | $91.27 \pm 1.1$ |
| **Best** | $90.56 \pm .11$ | $90.78 \pm .67$ | $91.51 \pm .04$ | $90.84 \pm .24$ | $90.91 \pm .62$ | $91.70 \pm .80$ |

Table 3: CIFAR-10 Results at a budget of 40 labeled examples, with batch size 10. Confidence intervals are standard errors based on three trials.

| | Test Accuracy | | | Pool Accuracy | | |
|---|---|---|---|---|---|---|
| | Pseudolabeling | UDA | FlexMatch | Pseudolabeling | UDA | FlexMatch |
| Confidence | $69.84 \pm .39$ | $70.42 \pm 1.1$ | $73.95 \pm .83$ | $\mathbf{70.09 \pm .60}$ | $70.19 \pm .90$ | $73.87 \pm .84$ |
| Entropy | $69.08 \pm .93$ | $69.77 \pm 1.3$ | $73.41 \pm .43$ | $68.75 \pm 1.4$ | $69.50 \pm .66$ | $73.47 \pm .56$ |
| Margin | $\mathbf{70.06 \pm .02}$ | $\mathbf{71.11 \pm 2.8}$ | $\mathbf{74.80 \pm .21}$ | $69.89 \pm 1.7$ | $\mathbf{71.28 \pm 2.6}$ | $\mathbf{74.94 \pm .16}$ |
| Coreset | $64.53 \pm 1.4$ | $63.94 \pm 2.4$ | $71.16 \pm 1.8$ | $64.13 \pm 1.1$ | $64.04 \pm 2.1$ | $70.87 \pm 1.6$ |
| BADGE | $68.82 \pm 2.1$ | $70.67 \pm 1.8$ | $74.68 \pm .23$ | $68.88 \pm 2.2$ | $70.77 \pm 1.9$ | $74.91 \pm .93$ |
| Random | $67.56 \pm 1.6$ | $69.39 \pm 2.7$ | $74.23 \pm .17$ | $67.67 \pm 1.3$ | $69.43 \pm 2.3$ | $74.07 \pm .65$ |
| **Best** | $70.06 \pm .02$ | $71.11 \pm 2.8$ | $74.80 \pm .21$ | $70.09 \pm .60$ | $71.28 \pm 2.6$ | $74.94 \pm .16$ |

Table 4: CIFAR-100 Results at a budget of 400 labeled examples, with batch size 100. Confidence intervals are standard errors based on three trials.

**F.6 Comparison Between Selection-Via-Proxy and Selection with Fine-tuning**

| | Test Accuracy | | | Pool Accuracy | | |
|---|---|---|---|---|---|---|
| | Fine-tune | Shallow Network | Linear Probe | Fine-tune | Shallow Network | Linear Probe |
| Confidence | 75.58 ± .08 | 75.29 ± .03 | **75.43 ± .04** | 81.11 ± .02 | 80.35 ± .01 | 80.46 ± .03 |
| Entropy | 74.95 ± .08 | 74.69 ± .06 | 74.91 ± .01 | 80.40 ± .11 | 79.76 ± .08 | 79.89 ± .02 |
| Margin | **75.65 ± .15** | **75.38 ± .06** | 75.38 ± .12 | **81.26 ± .03** | **80.44 ± .08** | **80.50 ± .06** |
| Coreset | 73.44 ± .05 | 73.31 ± .12 | 72.95 ± .18 | 78.14 ± .06 | 77.79 ± .02 | 77.51 ± .14 |
| BADGE | 75.49 ± .12 | 75.26 ± .12 | 75.37 ± .15 | 80.78 ± .05 | 80.20 ± .02 | 80.28 ± .04 |
| Random | 74.61 ± .15 | 74.61 ± .15 | 74.61 ± .15 | 78.64 ± .01 | 78.64 ± .01 | 78.64 ± .01 |
| **Best** | 75.65 ± .15 | 75.38 ± .06 | 75.43 ± .04 | 81.26 ± .03 | 80.44 ± .08 | 80.50 ± .06 |

Table 5: Selection-via-proxy results of ImageNet using CoCa ViT-B32. The results are evaluated with 400,000 labels. Confidence intervals are standard errors based on two trials.

| | Test Accuracy | | | Pool Accuracy | | |
|---|---|---|---|---|---|---|
| | Fine-tune | Shallow Network | Linear Probe | Fine-tune | Shallow Network | Linear Probe |
| Confidence | 97.84 ± .07 | 97.85 ± .05 | 97.86 ± .05 | 99.92 ± .02 | 99.67 ± .02 | 99.63 ± .02 |
| Entropy | 97.89 ± .08 | 97.87 ± .14 | 97.82 ± .06 | **99.93 ± .01** | 99.65 ± .02 | 99.61 ± .01 |
| Margin | **97.97 ± .12** | 97.88 ± .17 | 97.80 ± .03 | **99.93 ± .01** | **99.68 ± .01** | **99.64 ± .02** |
| Coreset | 97.79 ± .06 | 97.81 ± .19 | 97.77 ± .07 | 99.48 ± .02 | 98.94 ± .03 | 98.69 ± .03 |
| GALAXY | 97.94 ± .20 | **97.98 ± .12** | 97.84 ± .10 | 99.90 ± .01 | 99.66 ± .02 | 99.60 ± .02 |
| BADGE | 97.95 ± .08 | 97.84 ± .10 | **97.87 ± .06** | **99.93 ± .01** | 99.61 ± .02 | 99.58 ± .03 |
| BAIT | 97.87 ± .16 | 97.85 ± .14 | 97.84 ± .12 | 99.59 ± .04 | 99.32 ± .02 | 99.32 ± .02 |
| Random | 97.59 ± .22 | 97.59 ± .22 | 97.59 ± .22 | 98.18 ± .05 | 98.18 ± .05 | 98.18 ± .05 |
| **Best** | 97.97 ± .12 | 97.98 ± .10 | 97.87 ± .06 | 99.93 ± .01 | 99.68 ± .01 | 99.64 ± .02 |

Table 6: Selection-via-proxy results of CIFAR-10 using CLIP ViT-B32. The results are evaluated with 10,000 labels. Confidence intervals are standard errors based on four trials.

| | Test Accuracy | | | Pool Accuracy | | |
|---|---|---|---|---|---|---|
| | Fine-tune | Shallow Network | Linear Probe | Fine-tune | Shallow Network | Linear Probe |
| Confidence | **87.33 ± .26** | **86.37 ± .19** | **86.38 ± .17** | **93.75 ± .08** | **90.86 ± .03** | **90.90 ± .39** |
| Entropy | 86.89 ± .22 | 86.12 ± .18 | 86.14 ± .13 | 93.06 ± .05 | 90.56 ± .12 | 90.61 ± .05 |
| Margin | 87.30 ± .21 | 86.40 ± .36 | 86.68 ± .03 | 93.61 ± .03 | 90.76 ± .17 | 90.72 ± .05 |
| Coreset | 85.58 ± .28 | 85.08 ± .32 | 85.30 ± .38 | 89.41 ± .18 | 87.82 ± .50 | 87.63 ± .12 |
| GALAXY | 87.22 ± .20 | 86.28 ± .35 | 86.44 ± .24 | 93.05 ± .08 | 90.50 ± .02 | 90.64 ± .06 |
| BADGE | 87.20 ± .38 | 86.42 ± .23 | 86.55 ± .18 | 92.88 ± .15 | 90.16 ± .04 | 90.27 ± .19 |
| Random | 85.77 ± .20 | 85.77 ± .20 | 85.77 ± .20 | 87.72 ± .09 | 79.66 ± .03 | 79.66 ± .03 |
| **Best** | 87.33 ± .26 | 86.37 ± .19 | 86.38 ± .17 | 93.75 ± .08 | 90.86 ± .03 | 90.90 ± .39 |

Table 7: Selection-via-proxy results of CIFAR-100 using CLIP ViT-B32. The results are evaluated with 10,000 labels. Confidence intervals are standard errors based on four trials.

| | Test Macro F1 | | | Pool Macro F1 | | |
|---|---|---|---|---|---|---|
| | Fine-tune | Shallow Network | Linear Probe | Fine-tune | Shallow Network | Linear Probe |
| Confidence | 46.44 ± 2.14 | 48.89 ± 4.48 | 49.96 ± 3.87 | 62.40 ± .60 | 58.81 ± .94 | 59.93 ± 3.14 |
| Entropy | 50.00 ± .77 | 48.52 ± 8.04 | 50.54 ± 3.25 | **64.30 ± 2.24** | **65.47 ± 2.04** | 62.76 ± 1.85 |
| Margin | 50.55 ± 1.08 | 50.91 ± 1.80 | **52.26 ± 2.00** | 56.90 ± 3.17 | 56.52 ± 2.77 | 61.73 ± 1.63 |
| Coreset | 52.08 ± 1.71 | **53.13 ± 1.94** | 50.13 ± .77 | 49.33 ± 13.9 | 44.71 ± 11.6 | 38.71 ± .33 |
| GALAXY | **52.39 ± 3.48** | 49.87 ± 1.84 | 51.80 ± 3.85 | 62.41 ± 2.88 | 59.74 ± 1.54 | **62.87 ± 1.35** |
| BADGE | 49.88 ± 1.61 | 51.85 ± 0.82 | 50.51 ± 1.83 | 56.05 ± .52 | 54.31 ± 3.47 | 53.86 ± 1.52 |
| Random | 49.83 ± 1.26 | 49.83 ± 1.26 | 49.83 ± 1.26 | 38.47 ± .97 | 38.47 ± .97 | 38.47 ± .97 |
| **Best** | 52.39 ± 3.48 | 53.13 ± 1.94 | 52.26 ± 2.00 | 64.30 ± 2.24 | 65.47 ± 2.04 | 62.87 ± 1.35 |

Table 8: Selection-via-proxy results of iWildcam using CLIP ViT-B32. The results are evaluated with 21,000 labels. Confidence intervals are standard errors based on four trials.

## F.7 Alternative Semi-SL Methods with Selection-via-Proxy

In Table 10, 11, 12, 13, we assess how important the choice of Semi-SL algorithm is when we use SVP. We consider CIFAR10 and CIFAR100 for this comparison and for the proxy model consider Linear Probe. SoftMatch and FreeMatch are only evaluated on CIFAR10.

| | Test Accuracy | | | Pool Accuracy | | |
|---|---|---|---|---|---|---|
| | Fine-tune | Shallow Network | Linear Probe | Fine-tune | Shallow Network | Linear Probe |
| Confidence | $58.66 \pm .49$ | $57.82 \pm .37$ | $58.25 \pm .37$ | $72.47 \pm .32$ | $70.91 \pm .41$ | $71.42 \pm .27$ |
| Entropy | $58.14 \pm .75$ | $57.75 \pm .35$ | $58.02 \pm .29$ | $71.02 \pm 1.40$ | $70.87 \pm .27$ | $71.02 \pm .21$ |
| Margin | $59.51 \pm .37$ | $58.80 \pm .06$ | $58.98 \pm .30$ | $\mathbf{74.36 \pm .19}$ | $\mathbf{71.63 \pm .19}$ | $\mathbf{71.62 \pm .08}$ |
| Coreset | $57.71 \pm .26$ | $57.35 \pm .07$ | $56.75 \pm .69$ | $68.43 \pm .42$ | $66.50 \pm .40$ | $66.07 \pm .57$ |
| GALAXY | $59.41 \pm .22$ | $58.91 \pm .19$ | $59.10 \pm .28$ | $73.56 \pm .43$ | $71.32 \pm .76$ | $71.42 \pm .28$ |
| BADGE | $\mathbf{59.59 \pm .47}$ | $\mathbf{59.25 \pm .27}$ | $\mathbf{59.17 \pm .28}$ | $73.30 \pm .16$ | $70.92 \pm .05$ | $71.12 \pm .58$ |
| Random | $58.40 \pm .34$ | $58.40 \pm .34$ | $58.40 \pm .34$ | $68.46 \pm .13$ | $68.46 \pm .13$ | $68.46 \pm .13$ |
| **Best** | $59.59 \pm .47$ | $59.25 \pm .27$ | $59.17 \pm .28$ | $74.36 \pm .19$ | $71.63 \pm .19$ | $71.62 \pm .08$ |

Table 9: Selection-via-proxy results of fMoW using CLIP ViT-B32. The results are evaluated with 30,000 labels. Confidence intervals are standard errors based on four trials.

| | Pseudolabel | | UDA | | Flexmatch | |
|---|---|---|---|---|---|---|
| | Fine-tune | Linear Probe | Fine-tune | Linear Probe | Fine-tune | Linear Probe |
| Confidence | $97.72 \pm .06$ | $97.59 \pm .09$ | $\mathbf{97.93 \pm .08}$ | $97.74 \pm .04$ | $97.84 \pm .07$ | $\mathbf{97.86 \pm .05}$ |
| Entropy | $97.69 \pm .05$ | $97.60 \pm .04$ | $97.88 \pm .09$ | $\mathbf{97.80 \pm .03}$ | $97.89 \pm .08$ | $97.82 \pm .06$ |
| Margin | $97.82 \pm .05$ | $97.52 \pm .06$ | $97.77 \pm .05$ | $97.67 \pm .05$ | $\mathbf{97.97 \pm .12}$ | $97.80 \pm .03$ |
| Coreset | $97.38 \pm .08$ | $97.09 \pm .04$ | $97.48 \pm .03$ | $97.34 \pm .13$ | $97.79 \pm .06$ | $97.77 \pm .07$ |
| GALAXY | $\mathbf{97.87 \pm .06}$ | $97.55 \pm .08$ | $97.84 \pm .07$ | $97.74 \pm .05$ | $97.94 \pm .20$ | $97.84 \pm .10$ |
| BADGE | $97.75 \pm .08$ | $\mathbf{97.63 \pm .05}$ | $97.80 \pm .04$ | $97.70 \pm .05$ | $97.94 \pm .20$ | $97.87 \pm .06$ |
| BAIT | $97.70 \pm .04$ | $97.37 \pm .06$ | $97.68 \pm .05$ | $97.63 \pm .05$ | $97.87 \pm .16$ | $97.84 \pm .12$ |
| Random | $96.83 \pm .06$ | $96.75 \pm .11$ | $97.09 \pm .06$ | $97.31 \pm .07$ | $97.59 \pm .22$ | $97.59 \pm .22$ |
| **Best** | $97.87 \pm .06$ | $97.63 \pm .05$ | $97.93 \pm .08$ | $97.80 \pm .03$ | $97.97 \pm .12$ | $97.86 \pm .05$ |

| | SoftMatch | | FreeMatch | |
|---|---|---|---|---|
| | Fine-tune | Linear Probe | Fine-tune | Linear Probe |
| Confidence | $97.97 \pm .13$ | $97.85 \pm .03$ | $97.95 \pm .11$ | $97.84 \pm .13$ |
| Entropy | $\mathbf{97.99 \pm .14}$ | $\mathbf{97.91 \pm .13}$ | $97.87 \pm .12$ | $\mathbf{97.98 \pm .01}$ |
| Margin | $97.97 \pm .04$ | $97.85 \pm .01$ | $\mathbf{98.03 \pm .05}$ | $97.90 \pm .14$ |
| Coreset | $97.97 \pm .02$ | $97.64 \pm .07$ | $97.75 \pm .14$ | $97.70 \pm .05$ |
| GALAXY | $97.87 \pm .09$ | $97.77 \pm .01$ | $97.97 \pm .12$ | $97.85 \pm .10$ |
| BADGE | $97.94 \pm .11$ | $97.83 \pm .04$ | $97.88 \pm .09$ | $97.83 \pm .06$ |
| BAIT | $97.97 \pm .09$ | $97.77 \pm .14$ | $97.90 \pm .04$ | $97.85 \pm .02$ |
| Random | $97.57 \pm .05$ | $97.75 \pm .03$ | $97.58 \pm .09$ | $97.73 \pm .01$ |
| **Best** | $97.99 \pm .14$ | $97.91 \pm .13$ | $98.03 \pm .05$ | $97.98 \pm .01$ |

Table 10: **Test Accuracy**: Selection-via-proxy results of CIFAR10 using CLIP ViT-B32 for additional Semi-SL algorithms. The results are evaluated with 10,000 labels. Confidence intervals are standard errors based on four trials.

| | Pseudolabel | | UDA | | Flexmatch | |
|---|---|---|---|---|---|---|
| | Fine-tune | Linear Probe | Fine-tune | Linear Probe | Fine-tune | Linear Probe |
| Confidence | **99.92 ± .01** | 99.53 ± .03 | **99.93 ± .00** | 99.59 ± .01 | 99.92 ± .02 | 99.63 ± .02 |
| Entropy | **99.92 ± .01** | 99.54 ± .01 | **99.93 ± .00** | 99.59 ± .01 | **99.93 ± .01** | 99.61 ± .01 |
| Margin | 99.92 ± .00 | **99.57 ± .02** | **99.93 ± .00** | **99.61 ± .00** | **99.93 ± .01** | **99.64 ± .02** |
| Coreset | 99.37 ± .01 | 98.28 ± .01 | 99.40 ± .01 | 98.44 ± .04 | 99.48 ± .02 | 98.69 ± .03 |
| GALAXY | 99.90 ± .01 | 99.53 ± .01 | 99.91 ± .01 | 99.57 ± .01 | 99.90 ± .01 | 99.60 ± .02 |
| BADGE | 99.92 ± .00 | 99.53 ± .01 | 99.92 ± .00 | 99.54 ± .01 | **99.93 ± .01** | 99.58 ± .03 |
| BAIT | 99.45 ± .02 | 99.19 ± .01 | 99.55 ± .00 | 99.25 ± .01 | 99.59 ± .04 | 99.32 ± .02 |
| Random | 97.64 ± .04 | 97.67 ± .04 | 97.93 ± .03 | 97.93 ± .03 | 98.18 ± .05 | 98.18 ± .05 |
| **Best** | 99.92 ± .01 | 99.57 ± .02 | 99.93 ± .00 | 99.61 ± .00 | 99.93 ± .01 | 99.64 ± .02 |

| | SoftMatch | | FreeMatch | |
|---|---|---|---|---|
| | Fine-tune | Linear Probe | Fine-tune | Linear Probe |
| Confidence | **99.94 ± .00** | 99.61 ± .00 | **99.94 ± .00** | 99.64 ± .01 |
| Entropy | **99.94 ± .00** | **99.63 ± .02** | 99.92 ± .03 | 99.60 ± .01 |
| Margin | **99.94 ± .00** | **99.63 ± .00** | **99.94 ± .00** | **99.63 ± .01** |
| Coreset | 99.49 ± .00 | 98.65 ± .05 | 99.47 ± .03 | 98.64 ± .01 |
| GALAXY | 99.90 ± .00 | 99.59 ± .00 | 99.89 ± .01 | 99.60 ± .01 |
| BADGE | 99.92 ± .00 | 99.60 ± .00 | 99.93 ± .00 | 99.57 ± .00 |
| BAIT | 99.62 ± .03 | 99.28 ± .03 | 99.62 ± .04 | 99.29 ± .02 |
| Random | 98.22 ± .00 | 98.21 ± .02 | 98.21 ± .04 | 98.24 ± .00 |
| **Best** | 99.94 ± .00 | 99.63 ± .00 | 99.94 ± .00 | 99.63 ± .01 |

Table 11: **Pool Accuracy**: Selection-via-proxy results of CIFAR10 using CLIP ViT-B32 for different Semi-SL algorithms. The results are evaluated with 10,000 labels. Confidence intervals are standard errors based on four trials.

| | Peseudolabel | | UDA | | Flexmatch | |
| --- | --- | --- | --- | --- | --- | --- |
| | Fine-tune | Linear Probe | Fine-tune | Linear Probe | Fine-tune | Linear Probe |
| Confidence | 85.72 ± .08 | 84.96 ± .09 | 86.03 ± .14 | **89.84 ± .08** | **87.33 ± .26** | 86.38 ± .17 |
| Entropy | 85.72 ± .06 | 85.05 ± .12 | 85.94 ± .13 | 89.55 ± .08 | 86.89 ± .22 | 86.14 ± .13 |
| Margin | 85.66 ± .06 | 84.92 ± .17 | 86.17 ± .18 | 89.72 ± .05 | 87.30 ± .21 | **86.68 ± .03** |
| Coreset | 84.29 ± .19 | 83.80 ± .10 | 84.67 ± .20 | 86.35 ± .06 | 87.22 ± .20 | 86.44 ± .24 |
| GALAXY | **86.46 ± .20** | **85.18 ± .11** | **86.29 ± .06** | 89.78 ± .02 | 85.58 ± .28 | 85.30 ± .38 |
| BADGE | 85.79 ± .11 | 84.90 ± .10 | 86.24 ± .21 | 89.29 ± .02 | 87.20 ± .38 | 86.55 ± .18 |
| Random | 83.99 ± .17 | 83.92 ± .25 | 84.58 ± .11 | 86.66 ± .07 | 85.77 ± .20 | 85.77 ± .20 |
| **Best** | 86.46 ± .20 | 85.18 ± .11 | 86.29 ± .06 | 89.84 ± .08 | 87.33 ± .26 | 86.68 ± .03 |

Table 12: **Test Accuracy:** Selection-via-proxy results of CIFAR100 using CLIP ViT-B32. The results are evaluated with 10,000 labels. Confidence intervals are standard errors based on four trials.

| | Peseudolabel | | UDA | | Flexmatch | |
| --- | --- | --- | --- | --- | --- | --- |
| | Fine-tune | Linear Probe | Fine-tune | Linear Probe | Fine-tune | Linear Probe |
| Confidence | **92.32 ± .05** | **89.84 ± .08** | **92.71 ± .06** | 90.13 ± .03 | **93.75 ± .08** | **90.90 ± .39** |
| Entropy | 91.88 ± .07 | 89.55 ± .08 | 92.20 ± .11 | 89.93 ± .04 | 93.06 ± .05 | 90.61 ± .05 |
| Margin | 92.09 ± .05 | 89.72 ± .05 | 92.67 ± .05 | **91.80 ± .09** | 93.61 ± .03 | 90.72 ± .05 |
| Coreset | 88.40 ± .06 | 86.35 ± .06 | 88.67 ± .06 | 86.70 ± .09 | 89.41 ± .18 | 87.63 ± .12 |
| GALAXY | 91.78 ± .04 | 89.78 ± .02 | 92.03 ± .07 | 90.07 ± .07 | 93.05 ± .08 | 90.64 ± .06 |
| BADGE | 91.37 ± .09 | 89.29 ± .02 | 91.93 ± .02 | 89.63 ± .01 | 92.88 ± .15 | 90.27 ± .19 |
| Random | 86.75 ± .08 | 86.66 ± .07 | 87.06 ± .04 | 86.87 ± .03 | 87.72 ± .09 | 79.66 ± .03 |
| **Best** | 92.32 ± .05 | 89.84 ± .08 | 92.71 ± .06 | 90.18 ± .09 | 93.75 ± .08 | 90.90 ± .39 |

Table 13: **Pool Accuracy:** Selection-via-proxy results of CIFAR100 using CLIP ViT-B32. The results are evaluated with 10,000 labels. Confidence intervals are standard errors based on four trials.

