# OpenReview forum: "LabelBench: A Comprehensive Framework for Benchmarking Adaptive Label-Efficient Learning"
_DMLR — Accepted by DMLR_

### Review · Reviewer_eTL8 · 2023-12-17

**Recommendation:** 4
**Confidence:** 2

**Summary Of Contributions:**

The paper introduces LabelBench, a novel framework for benchmarking label-efficient learning techniques, combining active learning, semi-supervised learning, and fine-tuning of pretrained models. The framework's modular, open-sourced codebase invites community contributions, and its comprehensive experiments across various datasets highlight its broad applicability and potential in reducing resource and environmental impacts of machine learning training. The paper also introduces the "selection-via-proxy" approach, further optimizing the trade-off between training costs and model performance.

**Strengths:**

Strengths

1. Innovative Framework: LabelBench's integration of active learning, semi-supervised learning, and fine-tuning of pretrained models represents a significant advancement in label-efficient learning. This holistic approach allows for more nuanced and effective learning strategies.

2. Selection-Via-Proxy Approach: This novel approach optimizes data selection and labeling, striking a balance between reducing training costs and maintaining the advantages of active learning. It's a noteworthy contribution for resource optimization in model training.

3. Open-Sourced and Modular Codebase: The framework's modular design and open-source availability foster community engagement and contribution. This approach not only democratizes access to advanced learning techniques but also encourages collaborative improvements and innovations in the field.

**Audience:**

Yes

**Broader Impact Concerns:**

The authors have discussed Broader Impact Concerns.

**Claims And Evidence:**

The claims made in the submission are supported by accurate, convincing and clear evidence.

**Datasets And Benchmarks:**

The paper has the url code https://github.com/EfficientTraining/LabelBench and document.

**Extended Submissions:**

Not extended submissions.

**Limitations:**

See weakness.

**Requested Changes:**

Weak:

1. Typographical Error: There's a minor typographical error in the sentence: "Overall, LabelBench provides a light-weight benchmarking framework for researchers to test their algorithms on under more realistic and large-scale scenarios." The phrase "on under" should be corrected for clarity.

2. Limited Domain Scope in Experiments: The current scope of LabelBench's experiments is confined to image datasets. Expanding this to include domains like NLP and Biotechnology, especially considering the higher cost of labeling in these areas, could greatly enhance the framework's utility. Future work should consider integrating models like ProteinBERT (Bioinformatics) and Structure-aware protein self-supervised learning models (Bioinformatics) to address these domains.
[1] ProteinBERT: a universal deep-learning model of protein sequence and function. Bioinformatics.
[2] Structure-aware protein self-supervised learning. Bioinformatics.

3. Misleading Description of Benchmark Efficacy: The paper's claim that "Our benchmark demonstrates significantly better label-efficiencies than previously reported in active learning" and "yield state-of-the-art label-efficiency" could be misleading. As a benchmarking tool, LabelBench should primarily provide a platform for evaluation rather than claim efficacy in label efficiency, which is typically attributed to specific methods or algorithms. This distinction needs clarification to accurately reflect the purpose and capabilities of the benchmark.

4. Exploration of Initial Data Quality Impact: The dependence of LabelBench on the quality of initial data is a crucial aspect that warrants further investigation. Additional experiments focusing on how variations in initial data quality affect the overall performance and efficiency of the framework would be valuable. This exploration could provide deeper insights into the framework's robustness and adaptability in diverse real-world scenarios, where data quality is often variable.

**Strengths And Weaknesses:**

Strengths

1. Innovative Framework: LabelBench's integration of active learning, semi-supervised learning, and fine-tuning of pretrained models represents a significant advancement in label-efficient learning. This holistic approach allows for more nuanced and effective learning strategies.

2. Selection-Via-Proxy Approach: This novel approach optimizes data selection and labeling, striking a balance between reducing training costs and maintaining the advantages of active learning. It's a noteworthy contribution for resource optimization in model training.

3. Open-Sourced and Modular Codebase: The framework's modular design and open-source availability foster community engagement and contribution. This approach not only democratizes access to advanced learning techniques but also encourages collaborative improvements and innovations in the field.

Weak:

1. Typographical Error: There's a minor typographical error in the sentence: "Overall, LabelBench provides a light-weight benchmarking framework for researchers to test their algorithms on under more realistic and large-scale scenarios." The phrase "on under" should be corrected for clarity.

2. Limited Domain Scope in Experiments: The current scope of LabelBench's experiments is confined to image datasets. Expanding this to include domains like NLP and Biotechnology, especially considering the higher cost of labeling in these areas, could greatly enhance the framework's utility. Future work should consider integrating models like ProteinBERT (Bioinformatics) and Structure-aware protein self-supervised learning models (Bioinformatics) to address these domains.
[1] ProteinBERT: a universal deep-learning model of protein sequence and function. Bioinformatics.
[2] Structure-aware protein self-supervised learning. Bioinformatics.

3. Misleading Description of Benchmark Efficacy: The paper's claim that "Our benchmark demonstrates significantly better label-efficiencies than previously reported in active learning" and "yield state-of-the-art label-efficiency" could be misleading. As a benchmarking tool, LabelBench should primarily provide a platform for evaluation rather than claim efficacy in label efficiency, which is typically attributed to specific methods or algorithms. This distinction needs clarification to accurately reflect the purpose and capabilities of the benchmark.

4. Exploration of Initial Data Quality Impact: The dependence of LabelBench on the quality of initial data is a crucial aspect that warrants further investigation. Additional experiments focusing on how variations in initial data quality affect the overall performance and efficiency of the framework would be valuable. This exploration could provide deeper insights into the framework's robustness and adaptability in diverse real-world scenarios, where data quality is often variable.

---

### Review · Reviewer_UXps · 2023-12-28

**Recommendation:** 4
**Confidence:** 3

**Summary Of Contributions:**

This paper constructs a benchmarking framework for label-efficient learning methods, and incorporates scalable solutions to active learning methods for large-scale scenarios. Framework is comprehensive, modular, and aware of computational costs. I recommend accepting this work without hesitations.

**Strengths:**

1. This is the first framework, to the best of my knowledge, that combines semi-supervised learning, active learning, and various model fine-tuning methods such as linear probing, shallow probing, and full finetuning for large pre-trained models.
2. Codebase is very modular which will allow for easy incorporation of new methods.
3. They mitigate the computational cost by employing a variant of selection-via-proxy approach by Coleman et al. (2019) through (1) using a linear proble and shallow network models as potential proxies, and (2) precomputing and saving embeddings of each dataset.

**Audience:**

Yes

**Broader Impact Concerns:**

I do not have any concerns about the Broader Impact Statement. It includes sufficient detail.

**Claims And Evidence:**

Claims are accurate, convincing, and shows clear evidence.

**Datasets And Benchmarks:**

There is sufficient detail about all the aspects of data collection and organization.

**Extended Submissions:**

N/A

**Limitations:**

Framework is primarily focused on computer vision related tasks and models. It would be helpful to extend it to other domains such as natural language and graph data for both in-domain and out-of-domains taks.

**Requested Changes:**

I do not request any changes.

**Strengths And Weaknesses:**

Strengths:

1. This is the first framework, to the best of my knowledge, that combines semi-supervised learning, active learning, and various model fine-tuning methods such as linear probing, shallow probing, and full finetuning for large pre-trained models.
2. Codebase is very modular which will allow for easy incorporation of new methods.
3. They mitigate the computational cost by employing a variant of selection-via-proxy approach by Coleman et al. (2019) through (1) using a linear proble and shallow network models as potential proxies, and (2) precomputing and saving embeddings of each dataset.

The only weakness I see is that the framework is primarily focused on computer vision related tasks and models. It would be helpful to extend it to other domains such as natural language and graph data for both in-domain and out-of-domains taks.

---

### Review · Reviewer_8Mwn · 2024-01-13

**Recommendation:** 4
**Confidence:** 3

**Summary Of Contributions:**

The paper introduces LabelBench, a novel framework designed to benchmark and evaluate label-efficient learning techniques in machine learning. The key contributions and new knowledge presented by this submission are as follows:

Development of LabelBench: The authors developed LabelBench, a computationally efficient framework that enables the joint evaluation of multiple label-efficient learning techniques. This framework is particularly significant because it addresses a gap in existing benchmark and evaluation frameworks, which often do not consider the combined application of different label-efficient methods such as transfer learning, semi-supervised learning (Semi-SL), and active learning (AL).

Benchmark of State-of-the-Art Active Learning Methods: The paper showcases an application of LabelBench through a novel benchmark. This benchmark evaluates state-of-the-art active learning methods combined with semi-supervised learning for fine-tuning large pretrained vision transformers. The results demonstrate significantly improved label-efficiencies compared to previous active learning benchmarks.



Introduction of Selection-via-Proxy Technique: The paper proposes a 'selection-via-proxy' approach, enabling lightweight retraining schemes that substantially reduce the computational burden associated with retraining large-scale models during active learning. This approach involves using proxy models (like linear probes) during the data selection phase and only conducting end-to-end fine-tuning in the final stages. This technique was found to yield a ten-fold reduction in training costs.

Open-Source and Modular Codebase: The authors have made the LabelBench codebase open-source, encouraging broader community contributions. The modular design of the codebase allows easy integration and testing of new label-efficient learning methods and benchmarks.

Comprehensive Experiments and Evaluations: The paper presents extensive experiments benchmarking various deep active learning algorithms in combination with semi-supervised learning and large pretrained models. The experiments, which include evaluations on standard datasets like CIFAR-10, CIFAR-100, and ImageNet, as well as on more realistic datasets like iWildCam and fMoW, demonstrate the effectiveness of the LabelBench framework.

Insights into Label-Efficient Learning: The research findings provide valuable insights into the dynamics of label-efficient learning. It highlights the importance of combining active learning, semi-supervised learning, and large pretrained models, and offers an analysis of the effectiveness of different active learning strategies and semi-supervised learning techniques.

**Strengths:**

I mentioned it in the Strengths And Weaknesses section.

**Audience:**

Yes

**Claims And Evidence:**

Yes

**Datasets And Benchmarks:**

Yes

**Extended Submissions:**

No

**Limitations:**

Engineering Focus for a Comprehensive Benchmark: Since the paper does not propose a new learning approach but rather a benchmarking framework, it should thoroughly flesh out the engineering aspects. This involves ensuring the framework is versatile enough to integrate various learning paradigms and diverse datasets, including those from NLP.

Narrow Scope of Semi-Supervised Learning Methods: The paper currently utilizes a limited set of semi-supervised learning techniques. Expanding this to include a broader and more up-to-date range of semi-supervised methods.

Exclusion of Weakly Supervised Learning: The benchmark does not incorporate weakly supervised learning paradigms, which are crucial in scenarios where only noisy, limited, or imprecise supervision is available. Integrating weakly supervised learning would make the framework more versatile and reflective of real-world data challenges.

Limited Inclusion of NLP Datasets: The framework primarily focuses on image datasets and does not encompass NLP datasets. Including NLP datasets would not only broaden the scope of the benchmark but also test its applicability and robustness across different types of data modalities. Besides, focusing primarily on vision models and datasets could lead to methodologies and optimizations that are overly tailored to visual data, potentially limiting effectiveness when applied to other data types, such as textual data.


Addressing these limitations would significantly enhance the benchmark's scope, making it a more robust and comprehensive tool for evaluating label-efficient learning across a broader spectrum of machine learning applications.
I'd be happy to raise my score if the author addresses the issues I've raised.

============

The rebuttal has addressed my concerns. Thus, I decided to raise my score.

**Requested Changes:**

N/A

**Strengths And Weaknesses:**

**Strengths** :
Comprehensive Framework for Label-Efficient Learning: LabelBench offers a versatile and comprehensive framework that evaluates multiple label-efficient techniques (transfer learning, semi-supervised learning, and active learning) in conjunction. This is a significant advancement over existing frameworks that typically consider these methods in isolation.

Efficient Computational Approach: The introduction of the 'selection-via-proxy' technique for reducing computational costs is a notable innovation. This approach is particularly effective for large-scale models, making the benchmarking process more feasible and efficient.

Extensive and Robust Experiments: The paper presents thorough experimental evaluations across various datasets, including standard benchmarks and more complex, realistic datasets. This wide range of evaluations enhances the credibility and generalizability of the findings.

Open-Source and Modular Codebase: The open-source nature and modular design of the codebase encourage collaboration and facilitate easy integration of new methods or datasets, potentially accelerating further research in label-efficient learning.



**Weaknesses** :
Limited Scope in Learning Paradigms: The current framework focuses predominantly on transfer learning, semi-supervised learning, and active learning. However, it does not explicitly incorporate other learning paradigms such as weakly supervised learning. Integrating weakly supervised learning could enhance the framework's applicability, especially in scenarios where only noisy or indirect supervision is available.

You can refer to the following paper:

WRENCH: A Comprehensive Benchmark for Weak Supervision(https://arxiv.org/abs/2109.11377)

Limited Range of Semi-Supervised Learning Methods: The framework predominantly employs a narrow selection of semi-supervised learning techniques. While the methods used are valid, they represent only a small subset of the possibilities in this area. For a more comprehensive evaluation, incorporating a wider variety of semi-supervised learning methods, as highlighted in papers like "USB: A Unified Semi-supervised Learning Benchmark for Classification", would be beneficial. Testing newer and potentially more effective semi-supervised learning methods could significantly impact the framework's performance and generalizability.

You can refer to the following paper:

USB: A Unified Semi-supervised Learning Benchmark for Classification(https://arxiv.org/abs/2208.07204)

FreeMatch: Self-adaptive Thresholding for Semi-supervised Learning(https://arxiv.org/abs/2205.07246)

SoftMatch: Addressing the Quantity-Quality Trade-off in Semi-supervised Learning(https://arxiv.org/abs/2301.10921)



Restriction to Vision Models and Datasets: LabelBench primarily benchmarks vision models on image datasets. This focus limits its applicability in the broader context of machine learning, particularly in Natural Language Processing (NLP). Expanding the framework to include NLP datasets and models would greatly increase its utility and relevance in the field.

You can refer to the dataset of following project:

https://github.com/JieyuZ2/wrench

https://github.com/microsoft/Semi-supervised-learning